



# Importance of the forest state in estimating biomass losses from tropical forests: combining dynamic forest models and remote sensing

Ulrike Hiltner[1,2,3], Andreas Huth[1,4,5], Rico Fischer[1]

[1]Department of Ecological Modelling, Helmholtz-Centre for Environmental Research GmbH - UFZ, 04318 Leipzig, Germany

[2]Institute of Geography, Friedrich-Alexander-University Erlangen-Nuremberg, Erlangen, 91058 Germany

[3]Forest Ecology, Institute of Terrestrial Ecosystems, Department of Environmental Systems Science, ETH Zurich, 8092 Zurich, Switzerland

[4]German Centre for Integrative Biodiversity Research – iDiv Halle-Jena-Leipzig, 04103 Leipzig, Germany

[5] Institute for Environmental Systems Research, University Osnabruck, 49076 Osnabruck, Germany

Correspondence to Ulrike Hiltner (u.hiltner.uh@gmail.com)

**Abstract**. Disturbances, such as extreme weather events, fires, floods, and biotic agents, can have strong impacts on the dynamics and structures of tropical forests. In the future, the intensity of disturbances will likely further increase, which may have more serious consequences for tropical forests than those we have already observed. Thus, quantifying aboveground biomass loss of forest stands due to tree mortality (hereafter biomass loss) is important for the estimation of the role of tropical

forests in the global carbon cycle. So far, the long-term impacts of altered tree mortality on rates of biomass loss have been described little.

This study aims to analyse the consequences of long-term elevated tree mortality rates on forest dynamics and biomass loss. We applied an individual-based forest model and investigated the impacts of permanently increased tree mortality rates on the growth dynamics of humid, terra firme forests in French Guiana. Here, we focused on biomass, leaf area index (LAI), forest

height, productivity, forest age, quadratic mean stem diameter, and biomass loss. Based on the simulations, we developed a multiple linear regression model to estimate biomass losses of forests in different successional states from the various forest attributes.

The findings of our simulation study indicated that increased tree mortality altered the succession patterns of forests in favour of fast-growing species, which changed the forests' gross primary production, though net primary production remained stable.

Tree mortality intensity had a strong influence on the functional species composition and tree size distribution, which led to lower values in LAI, biomass, and forest height at the ecosystem level. We observed a strong influence of a change in tree mortality on biomass loss. Assuming a doubling of tree mortality, biomass loss increased (from 3.2% y⁻¹ to 4.5% y⁻¹). We also obtained a multidimensional relationship that allowed for the estimation of biomass loss from forest height and LAI. Via an example, we applied this relationship to remote sensing data of LAI and forest height and mapped biomass loss for French

Guiana. We estimated a mean biomass loss rate of 3.2% per year.





The approach described here provides a novel methodology for quantifying biomass loss, taking the successional state of tropical forests into account. Quantifying biomass loss rates may help to reduce uncertainties in the analysis of the global carbon cycle.

**Keywords**. FORMIND forest model, MODIS MCD15A2H Version 6 LAI, Lidar, forest height map, French Guiana,
biomass loss map, biomass turnover time

## 1. Introduction

Tropical forests represent an important pool in the global carbon cycle, as they store approximately 55% of the amount of global forest carbon (471 ± 93 PgC) in their living biomass  (Pan et al., 2011). Intact tropical forests assimilate an average of 0.96 ± 0.46 PgC of carbon per year (Hubau et al., 2020). This carbon sink behaviour of tropical forests has considerably
reduced the growth rate of atmospheric carbon dioxide (Friedlingstein et al., 2019; Le Quéré et al., 2016). However, the carbon assimilation capacity of forests is affected by tree mortality due to disturbances, which can cause rapid, extensive carbon loss (Chambers et al., 2013; Fisher et al., 2008; Korner, 2003; Pugh et al., 2019; Seidl et al., 2014). Increased tree mortality due to disturbances has been related to a reduction in the carbon sink of tropical forests (Brienen et al., 2015; Hubau et al., 2020). A number of studies have discussed different climate-controlled mortality drivers, such as temperature (Clark et al., 2010), vapour
pressure deficit (Trenberth et al., 2014), drought (Fauset et al., 2019; Phillips et al., 2010), and wind-throw (Chambers et al., 2009; Magnabosco Marra et al., 2016; Marra et al., 2014; Negrón-Juárez et al., 2010, 2018; Rifai et al., 2016; Silvério et al., 2019). In addition, mechanical disturbances, such as insect calamities (Coley and Kursar, 2014), fires (Barlow et al., 2003; Brando et al., 2014; Slik et al., 2010), and lianas (Ingwell et al., 2010; Nepstad et al., 2007a; Wright et al., 2015), may also lead to increased tree mortality. The expected increase in the frequency and intensity of those disturbances may result in an
overall increase in tree mortality and its associated physiological mechanisms (McDowell et al., 2018). Higher levels of tree mortality thus present a major risk to climate mitigation efforts (e.g., REDD+: Reducing Emissions from Deforestation and Forest Degradation), as reductions in carbon assimilation rates and a decrease in the carbon stocks of tropical forests could counteract attempts to compensate for climate change (Gumpenberger et al., 2010; Körner, 2017; Le Page et al., 2013).

Mortality is a complex process because the causes leading to tree death can be diverse. Trees can die naturally from senescence
or from forest disturbances which may be abrupt or continuous and may have abiotic or biotic, allogenic or autogenic, as well as extrinsic or intrinsic causes (Franklin et al., 1987; McDowell et al., 2018). Furthermore, drivers of tree mortality often occur in combination, so the primary factors of death are not obvious (Franklin et al., 1987; McDowell et al., 2018). Tree mortality leads to temporal changes in stand structure, tree species composition, and releases of resources, particularly biomass (Franklin et al., 1987; Hülsmann et al., 2018). Consequently, tree death affects important forest growth processes, including tree growth
and establishment, which are influenced by species-specific competition strategies (Snell et al., 2014) as well as by environmental and competitive factors such as light availability (Kuptz et al., 2010; Poorter, 1999; Uriarte et al., 2004). The influence of tree mortality on forest growth dynamics is determined by the disturbance intensity, which can range from the



temporary loss of vitality to the mortality (Kindig and Stoddart, 2003) of individual trees, forest stands, and entire landscapes. Finally, tree mortality events are heterogeneously distributed such that spatial patterns can be scattered or clustered (Franklin

et al., 1987). Empirical studies have already analysed the effects of short-term disturbances (i.e., intra-annual or over a few years) on increases in tropical tree mortality (e.g., Barlow et al., 2003; Brando et al., 2014; Chambers et al., 2009, 2013; Doughty et al., 2015a; Holzwarth et al., 2013; Magnabosco Marra et al., 2016; Marra et al., 2014; McDowell et al., 2018; Negrón-Juárez et al., 2010, 2017; Nepstad et al., 2007b; Phillips and Brienen, 2017; Slik et al., 2010; Stovall et al., 2019; Wright et al., 2015). Nevertheless, using empirical studies that are limited in space and time, it is difficult to quantify the long-

term effects of permanently increased tree mortality levels and to assess the consequences of such alterations on the dynamics, the structures, and the successional states of forests. Also, new remote sensing technologies offer enhanced potential for measuring the vertical and horizontal structures of forests at country to global scales (e.g., Bi et al., 2015; Hall et al., 2011; Lefsky et al., 2002, 2005; Myneni et al., 2015; Simard et al., 2011; Le Toan et al., 2011). Remote sensing products have previously been used for large-scale identification of tree mortality following disturbances (e.g., Pugh et al., 2019; Senf and

Seidl, 2020); however, the estimation of biomass loss due to tree mortality for forests at different states remains still uncertain.

In this context, individual-based forest gap models offer an approach by which to analyse forest dynamics (Botkin et al., 1972; Bugmann, 2001; Bugmann et al., 2019; Fischer et al., 2016; Shugart, 2002). Individual-based forest models are parameterised with forest inventory data to allow for the investigation of forest growth dynamics over longer periods. By simulating the growth, establishment, mortality, and competition among trees within a forest, these models can contribute in estimating the

biomass gain and loss of tropical forests (e.g., Hiltner et al., 2018, 2021; Maréchaux and Chave, 2017). As a result of gap formation after tree falling (Fischer et al., 2016; Huth et al., 1998), simulation areas consist of a mosaic of forest stands on which the vertical and horizontal structures and dynamics of forests in different successional states are modelled (Botkin, 1993; Botkin et al., 1972; Bugmann, 2001; Fischer et al., 2016; Shugart, 1984). Structural state variables describing successional states of forests, such as tree size distributions and functional tree species compositions, play a major role in the estimation of

the carbon budgets of forest stands and entire landscapes (Bohn and Huth, 2017; Fischer et al., 2018, 2019; Rödig et al., 2017, 2018a, 2019; Rüger et al., 2020). Successional state variables of forests can be derived on large spatial scales (e.g., country to global levels) through a combination of individual-based forest gap modelling and remote sensing (Rödig et al., 2017, 2019; Shugart et al., 2015, 2018), as this allows for a quantification of the spatial variation in forest structure due to tree mortality (Rödig et al., 2017). The combination of individual-based forest gap models and remote sensing methods may also provide

information on the spatial distribution of the annual rates of aboveground biomass loss due to tree mortality (hereafter biomass loss).

The aims of this study are to investigate the impacts of permanently increased tree mortality rates on forest dynamics, to provide a framework for estimating biomass loss in terra firme forests at different successional states, and to derive a sample map of biomass loss for an entire country (i.e., French Guiana). Here, we address the following research questions in detail:




1.  What are the consequences of permanently increased tree mortality levels on the dynamics of forest attributes (e.g., aboveground biomass, forest height, gross primary production, net primary production, leaf area index, quadratic mean stem diameter, mean forest age, and biomass loss) in tropical forests?

2.  Can the biomass loss of tropical forests be estimated using various forest attributes that can be derived from remote sensing?

## 2. Materials and Methods

We applied the 'terra firme' version of the dynamic individual-based forest model FORMIND (Fischer et al., 2016; Hiltner et al., 2018; Köhler and Huth, 2004) and simulated the effects of long-term increased tree mortality levels on the dynamics of multiple forest attributes (Figure 1). We included aboveground biomass (hereafter biomass), mean forest height, gross primary production (GPP), net primary production (NPP), leaf area index (LAI), biomass turnover time ($\tau_B$), quadratic mean stem diameter (QMD), mean forest age, and biomass loss ($m_{AGB}$) in our assessment. We analysed all of these forest stand attributes in relation to the intensity of increased tree mortality. Each simulated forest stand used in the analysis has the area of one hectare, with the forest states of each hectare differing from each other in each simulated time step and scenario. Then, we developed a multiple linear regression model by testing different forest state attributes, such as LAI and forest height, as proxy variables. In addition, we derived a sample map for the biomass loss and biomass residence time of an entire region by using values for forest height and LAI obtained from satellite products. Simulated terra firme forests of French Guiana served as a case study.

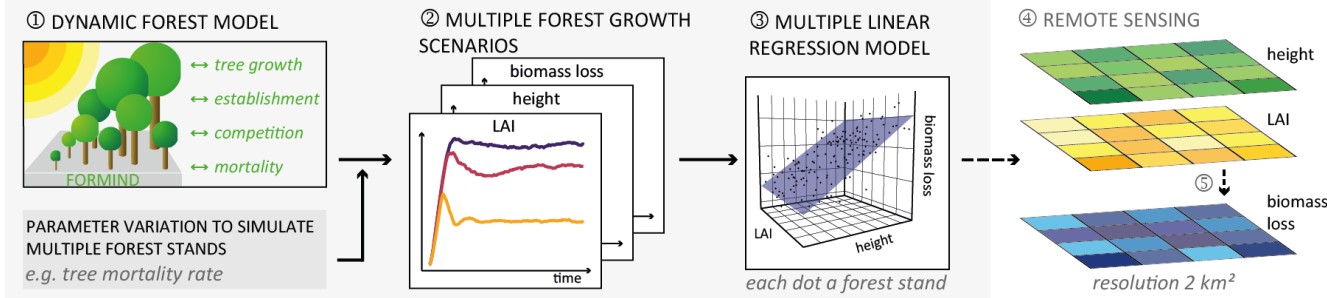

**Figure 1: Framework developed for estimating biomass loss by linking a dynamic forest model and remote sensing. 1. A forest model was applied to (2.) simulate the succession dynamics of forest stands of various forest attributes, such as LAI, forest height, and biomass loss, in a set of different tree mortality scenarios (results used to answer research question 1). A simulated forest stand has the area of one hectare, with the forest states of each simulated hectare differing at each simulation time step and scenario. 3. Then, we developed a multiple linear regression model to the simulated forest states with LAI and forest height as proxy variables and biomass loss as the response (results to answer research question 2). 4. In addition, we applied the multiple linear regression model to remote sensing maps containing the values of the investigated forest attributes (LAI and forest height) to (5.) derive a sample map of biomass loss.**



## 2.1 Study region

The study region is French Guiana, 95% of which is covered by humid, lowland terra firme forests (Hammond, 2005; Stach et al., 2009). These forests are characteristic for the Guiana Shield (Grau et al., 2017). The forests are generally species-rich, with an average of 150 to 200 tree species per hectare (Gourlet-Fleury et al., 2004), and are dense in biomass stock (Johnson
et al., 2016; Rödig et al., 2017; Saatchi et al., 2011).

## 2.2 Forest model FORMIND

### 2.2.1 Model description

To analyse the forest dynamics under the impacts of different levels of disturbance, we applied the 'terra firme parameterisation' of the forest model FORMIND v3.2 (Fischer et al., 2016) and took relevant parameter values from Hiltner
et al. (2018), including tree growth, mortality, and establishment (see Supplements Tab. S1 and Tab. S2). FORMIND is an individual-based forest gap model that describes forest dynamics, tree growth, and changes in forest structures on a simulation area (1 hectare to multiple square kilometres) consisting of 20 m by 20 m patches that interact with each other (see Fig. S1), where trees are not positioned explicitly. Our assumption in the 'terra firme parameterisation' of FORMIND is that forest stands also have no explicit position.

Every tree with a stem diameter at breast height (DBH) ≥ 0.1 m was simulated considering the following main processes at annual time steps: tree growth, establishment, mortality, and competition for light and space. The biomass gain of a tree results from the difference between photosynthetic production and respiratory losses (Fischer et al., 2016; Hiltner et al., 2018, 2021). In the model, tree mortality is a key driver of forest dynamics. Tree mortality increases if the space for canopy expansion is limited, which depends on a tree's position within the forest stand (self-thinning by crowding), whether tree growth is reduced
(growth-dependent), and whether surrounding trees die after large trees fall (gap formation). Finally, each tree is subject to a tree mortality rate, which is stochastic. Here, we modified the tree mortality rate (eq. 1) to induce heterogeneity in the horizontal and vertical forest structures (i.e., tree size distribution and functional species composition) of terra firme forests in French Guiana. Our study does not focus on short-term disturbances, but on the effects of long-term changes (> 100 years) in the intensity of tree mortality. Possible factors altering tree mortality rates in the forest model include environmental drivers, such
as extreme climate events, forest fires, wind-throw, and diseases. The tree mortality rate refers to individual trees at the stand level and is not stand-replacing.

For the generic forest model parameterization of French Guiana's terra firme forests, tree species were classified into eight plant functional types (PFTs) according to species-specific traits, i.e., the maximum incremental rates of DBH and maximum tree height. We assume here that major parts of the terra firme forests can be characterised on the basis of three functional
species groups: light-requiring species, species with intermediate light requirements, and shade-tolerant species. This functional species diversity is considered to be sufficient to capture forest succession dynamics in tropical forests (Fischer et





al., 2018; Rödig et al., 2017; Rüger et al., 2020). Detailed model descriptions can be found in Fischer et al. (2016), in Hiltner et al. (2018), and online at www.formind.org.

### 2.2.2 Simulation settings

To investigate the effects of different tree mortality intensities on the dynamics and the structure of terra firme forests, we developed seven simulation scenarios: a baseline scenario and six scenarios with permanently altered tree mortality rates (Tab. 1). The baseline scenario based on observed tree mortality rates ($m_{bl}$), which were, on average, 1.29%. This was computed by averaging the species-specific tree mortality rates of all PFTs (Tab. S2). To obtain tree mortality rates for all scenarios, the baseline's tree mortality rate ($m_{p.bl}$) was multiplied by a factor ($f$) for each scenario ($sc$), resulting in the following equation:

$$m_{p,sc} = f \cdot m_{p,bl}, \qquad \text{with } f \in \left\{\frac{1}{4}, \frac{1}{3}, \frac{1}{2}, 2, 3, 4\right\}. \qquad (1)$$

This resulted in different forest disturbance intensities at the stand level per simulated scenario (Tab. 1). The scenario with $f = 1$ represented the baseline scenario. In this study, we simulated forest stands with an area of one hectare and consisting of interacting patches on which the forests grow (patch size 20 m · 20 m). The one-hectare stands extended over a total simulation area of 16 hectare per scenario. Forest dynamics were computed in an annual time step that started in year 0 on bare ground

and ended after 300 years. In the baseline scenario, a forest stand reached equilibrium after 210 years.

**Table 1: Average tree mortality intensities per simulation scenario plus specification (see eq. 1).**

| Factor ($f$) | Average tree mortality rate ($m_{p,sc}$) in a$^{-1}$ | Specification |
|---|---|---|
| 1/4 | 0.003225 | Low impact |
| 1/3 | 0.0042957 | |
| 1/2 | 0.00645 | |
| 1 | 0.0129 | Baseline |
| 2 | 0.0258 | |
| 3 | 0.0387 | |
| 4 | 0.0516 | High impact |

From the model outputs of all scenarios, we analysed the average development of multiple forest attributes (averaged over 16 hectares), such as aboveground biomass (AGB), LAI, forest height (mean height of the tallest three trees per 40 m · 40 m;

Rödig et al., 2017; Simard et al., 2011), gross primary production, net primary production, quadratic mean stem diameter (square root of the sum of squared stem diameters per tree divided by the number of trees in a stand), mean forest age (arithmetic mean age of the 25 oldest trees per simulated 1 hectare area, selecting the oldest tree per patch), and biomass loss ($m_{AGB}$), which we defined as annual proportion of dead biomass ($AGB_{dead}$) to total stand biomass ($AGB_{total}$):

$$m_{AGB} = AGB_{dead} \cdot AGB_{total}^{-1}. \qquad (2)$$





In our forest model, we use a non-linear relationship between the $AGB_t$ of tree ($t$) and its stem diameter ($D_t$):

$$AGB_t = \frac{\pi}{4} \cdot D_t^2 \cdot H_t \cdot F_t \cdot \frac{\rho_t}{\sigma_t}, \qquad (3)$$

where $H_t$ is the tree height, $F_t$ is a form factor, $\rho_t$ is the wood density, and $\sigma_t$ is the fraction of aboveground biomass attributed to the stem (Fischer et al., 2016). Then, the tree biomasses are summed up to yield the total biomass, $AGB_{total}$.

In addition, we computed the time period over which each forest attribute reached the stable state (hereafter: equilibrium time) as well as the mean biomass turnover times ($\tau_B$) with $\tau_B$ averaged over all successional states (simulated years 0 – 300). According to Carvalhais et al. (2014), turnover times can be defined as the ratio of the biomass stock to the flux (i.e., influx or outflux) of biomass. However, biomass outflux is not yet observable over large spatial scales (Thurner et al., 2016). Therefore, it was defined that biomass outflux equals biomass influx for forests in equilibrium (Carvalhais et al., 2014). Transferred to our study, the stock corresponds to the total biomass, influx to NPP, and outflux to dead biomass. Therefore, the following holds true for forests in equilibrium:

$$\tau_B = stock \cdot flux^{-1} = AGB_{total} \cdot NPP^{-1} = AGB_{total} \cdot AGB_{dead}^{-1} . \quad (4)$$

The $\tau_B$ value can also be calculated from eqs. 2 and 4 as the reciprocal of the biomass loss:

$$\tau_B = 1 \cdot m_{AGB}^{-1} \quad \text{for } m_{AGB} > 0. \qquad (5)$$

Using eq. 4, we calculated $\tau_B$ by taking forest succession into account.

**2.3    Derivation of a multiple linear regression model to estimate biomass loss**

To estimate the biomass loss, we analysed a number of forest simulations which produced a large number of different forest stands (each 1 ha on 16 ha simulation areas) in different successional states (per simulated year) with unique functional species compositions and tree size distributions. Thus, we generated a total of 33,600 terra firme forest stands. We assumed that biomass loss can be related to other forest attributes (e.g., biomass, LAI, forest productivity, and forest height). For a multiple linear regression model, the temporal and spatial components are not important since forest states are considered independently of either.

We tested different statistical models using different combinations of the proxy variables. The best estimate was made by a multiple linear regression model which describes variations in $m_{AGB}$ as a function of two proxy variables, LAI and forest height (Tab. S3). We estimated $m_{AGB}$ (in units of y$^{-1}$) as follows:

$$m_{AGB} = \beta_H \cdot H + \beta_L \cdot L + \varepsilon, \qquad (6)$$

where $H$ is the forest height, $L$ is the LAI (-), $\varepsilon$ is the error term, and $\beta_i$ are the regression coefficients of the $i^{th}$ forest attributes. The intercept was set to zero, as the biomass loss is expected to be zero when both LAI and forest height equal zero.





To test whether the linear regression model is robust, we simulated additional scenarios with altered productivity rates. Based on this new data together with the previous data, we fitted an alternative multiple linear regression model. Similarity between

both linear multiple regression models implied high robustness of the original model. For further information, see the supplemental material (Fig. S8 and Fig. S9).

## 2.4 Estimation of the country-wide biomass loss

### 2.4.1 Input maps

To estimate forest height, we used a global map in the WGS-84 geographical projection with a pixel size of approximately one

kilometre (Simard et al., 2011; Fig. S4.a). To create an LAI map, we used 139 data layers from the MCD15A2H Version 6 Moderate Resolution Imaging Spectroradiometer (MODIS) Level 4 with a pixel size of 500 m and averaged the LAI values between 2004-01-31 and 2006-12-31 to reduce the overall LAI variance (Myneni et al., 2015). We harmonised and stacked the two input maps by first projecting the LAI map onto the coordinate reference system of the forest height map using the Geospatial Data Abstraction Library for French Guiana (www.gdal.org). Resampling was conducted with the bilinear method.

The spatial aggregation of the LAI map (Fig. S4.b) was performed by calculating the mean values of pixels whose centres were within 1-kilometre cells of the forest height map.

### 2.4.2 Output maps

The biomass loss ratio was estimated for each pixel by applying the multiple linear regression model (eq. 6) to the two input maps (see Fig. S4). We compared the density distributions of both input data sets with the ranges of FORMIND's simulation

results (LAI and forest height (H)). No correction factors were required for the extrapolations (see Fig. S7). The biomass loss values were then averaged over a pixel size of 2 km$^2$. We simulated forest stands of one hectare with the forest model. This fine resolution allowed us to scale up to the accuracy of remote sensing products. We used a resolution of 2 km for the final biomass loss map, although the input data are available in 0.5 km (LAI, Myneni et al., 2015) and 1 km (forest height, Simard et al., 2011). Our regression model estimated negative biomass loss ratios for a small portion of pixels, which were excluded

from the biomass loss map. This was mainly the case for pixels without forest cover, according to a land use map (see Fig. S6) published by Stach et al. (2009). To create the biomass turnover time map, we computed the reciprocal of each pixel for the biomass loss map (see eq. 4).

We tested the reliability of the mapped biomass loss in the underlying input maps for the LAI and forest height via a sensitivity analysis regarding variations of ±30% as compared to the original input maps. This resulted in four new input maps (LAI$_{+30\%}$,

H$_{+30\%}$, LAI$_{-30\%}$, H$_{-30\%}$). We then applied the multiple linear regression model (see eq. 6, Tab. 2) to all possible pair-wise combinations of these four new input maps (i.e., LAI$_{+30\%}$-H$_{+30\%}$, LAI$_{-30\%}$- H$_{-30\%}$, LAI$_{+30\%}$-H$_{-30\%}$, LAI$_{-30\%}$- H$_{+30\%}$) to obtain four uncertainty maps. We calculated $\Delta m_{AGB}$ per pixel, which is defined as the difference between each of these four new maps and the biomass loss map obtained from the original input maps. Thus, $\Delta m_{AGB}$ represents the variation in biomass loss given 30%





variation in the input variables. Furthermore, we compared our map with forest plot data, provided by Brienen et al. (2015),
and with map data of Johnson et al. (2016). Please refer to the supplementary material for details on the computer software
used in this study.

## 3    Results

### 3.1    Influence of increased tree mortality on forest succession dynamics

To analyse the influences of varying tree mortality intensities, we simulated succession dynamics, which were affected by
competition among individual tree species belonging to species groups. Here, we show that successional stages can be
differentiated based on the development of the total stand biomass (Fig. 2). After 40 years of forest succession, the simulated
stand biomass peaked at 500 $t_{ODM}$ ha$^{-1}$. This peak in stand biomass was caused by a high GPP of the pioneer species (GPP$_{pioneer}$
= 83 $t_{ODM}$ ha$^{-1}$; Fig. S1.a). After the early successional stage (years 0 – 40), the stand biomass fell slightly until year 100  due
to the rapidly declining pioneer biomass, while the biomasses of other species increased (mid successional stage; Fig. 2). After
100 years, the stand biomass stabilised at approximately 420 $t_{ODM}$ ha$^{-1}$a$^{-1}$ (average over years 100–300), while the functional
species composition reached a steady state after only 210 years (Fig. 2). In the late successional stage (gap dynamics), climax
species and species with intermediate light requirements fixed five times more carbon in biomass than pioneer species (NPP
of baseline scenario; Fig. S2.b).

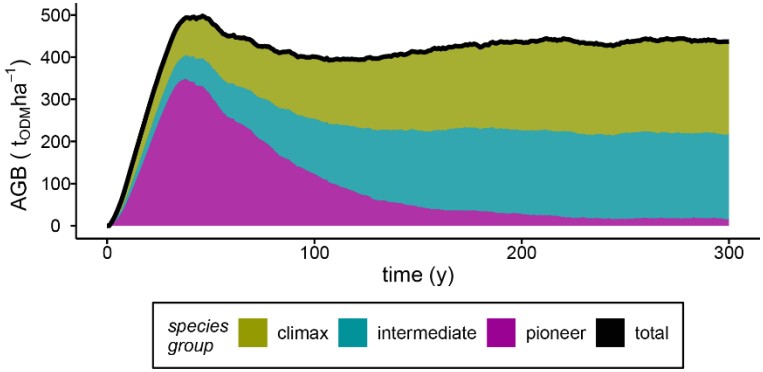

**Figure 2: The baseline scenario's aboveground biomass (AGB) per species group and the total biomass for simulations of terra firme
forests. (ODM: organic dry matter).**

Our simulation results reveal a sensitive response of biomass loss to increased tree mortality intensities (Fig. 3.a). At higher
tree mortality levels, higher biomass loss with greater variance emerged. At the level with the highest tree mortality, a peak in
the biomass loss rates occurred at approximately 0.12 y$^{-1}$ during the early phase of forest succession before levelling off at a
value of 0.08 y$^{-1}$ in the steady state (Fig. 3.a). Due to the higher biomass loss, the light climate in the forest stand changed (Fig.
3.c). The pioneer species were able to establish quickly in forest gaps. Hence, the GPP of the pioneer species was highest
among all species groups (Fig. S2.a), which also affected the productivity of the total stand (Fig. 3.e – 3f, Fig. 4.a). Despite



the distinctly higher GPP values obtained in the case of higher biomass loss rates, NPP did not change distinctly among the different scenarios (Fig. 3.f). Thus, the tree mortality intensity had a strong influence on the species composition (e.g., higher

pioneer GPPs; Fig. S2.a) and forest structure (e.g., QMD and mean stand age; Fig. S3), which led to lower LAI, biomass, and mean forest height values at the ecosystem level than those of the reference (Fig. 3.b – 3.d). In addition, structural changes gave rise to modified forest stand dynamics, with unique succession patterns depending on the intensity of the disturbance due to tree mortality.

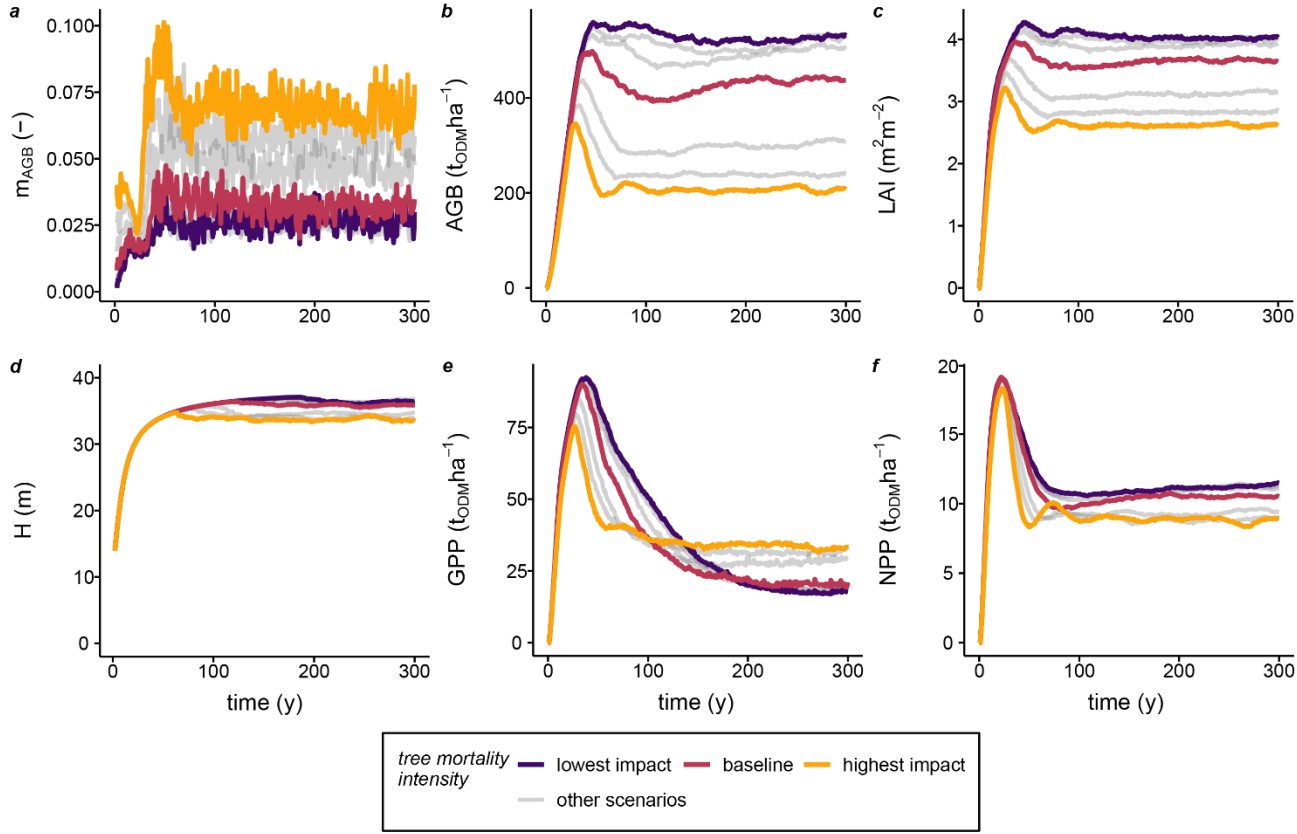


**Figure 3: Simulation results of the (a) biomass loss (m$_{AGB}$), (b) aboveground biomass (AGB), (c) leaf area index (LAI), (d) mean forest height (H), (e) gross primary production (GPP), and (f) net primary production (NPP) of terra firme forest stands under different tree mortality intensities. Grey lines indicate the entire set of scenarios under varying tree mortality rates (eq. 1) (ODM: organic dry matter; for further details see Fig. S2 and Fig. S3).**

Furthermore, we analysed how the tree mortality level affected the time needed to reach equilibrium (Fig. 4.b). GPP responded particularly sensitively and inversely proportionally to the tree mortality intensity, showing a strong decrease with rising tree mortality levels. In contrast, other forest attributes, such as biomass and NPP, had altogether shorter equilibrium times than that of GPP, responding inversely proportionally to the tree mortality level.



Finally, we evaluated the effect of increasing tree mortality rates on the turnover time of biomass (eq. 5) in the forest stands

while taking forest succession into account (Fig. 4.c). The biomass turnover time $\tau_B$ was more than halved at a four-fold higher tree mortality intensity as compared to the baseline ($\tau_{B,(f=1)} = 34$ y; $sd_{B,(f=1)} = 12$ y; $\tau_{B,(f=4)} = 46$ y, $sd_{B,(f=4)} = 43$ y; $\tau_{B,(f=1/4)} = 15$ y, $sd_{B,(f=1/4)} = 5$ y). Important forest properties are profoundly affected if the functional species composition, tree size distribution, or forest dynamics are changed.

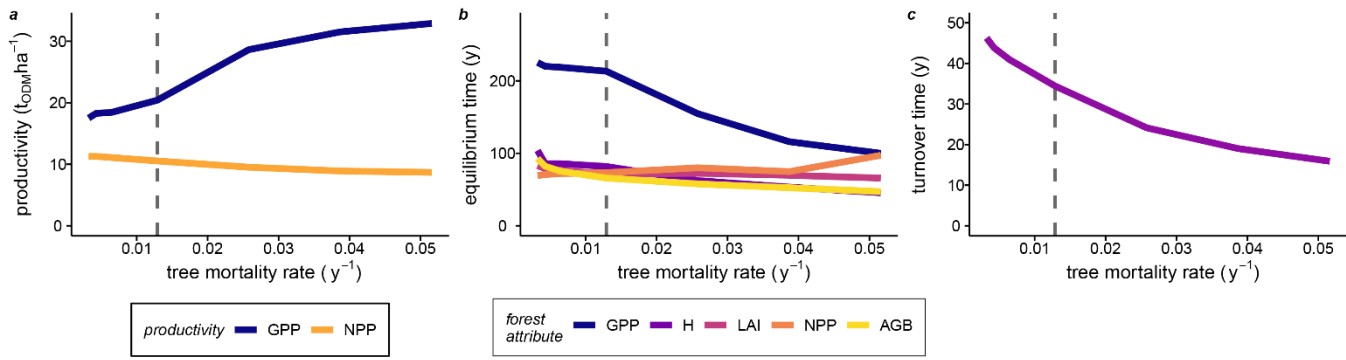

**Figure 4: Influence of tree mortality levels in simulated terra firme forests on (a) mature forests' mean GPP and mean NPP (averages over simulation years 250 – 300), (b) the time until the forest attributes reached equilibrium states, and (c) the mean biomass turnover times, represented as the reciprocal values of biomass loss (cf. eq. 5; averages over years 0–300). Dashed lines indicate the baseline scenario (GPP: gross primary production, NPP: net primary production, H: forest height, LAI: leaf area index, AGB: aboveground biomass, ODM: organic dry matter).**

### 3.2    Estimation of biomass loss using single and multiple forest attributes

In a further analysis, we assessed how biomass loss can be derived from different proxy variables, such as the mean forest height and LAI. Including forests at different successional states, we tested the relationships between several single forest attributes and biomass loss but did not find distinct relationships (Fig. 5.a – 5.c; Tab. S3: regression model types 2 – 6). The

biomass loss showed a widely scattered range of values and thus unclear relationships to all single forest attributes during the early successional stage (forest age < 20 years; Fig. 5.a – 5.c). For instance, the LAI values of less-disturbed, old-growth forests (i.e., LAI = 4 during the gap dynamics stage) indicated similar biomass losses to forests in the early stages of succession. Relationships between single forest attributes and biomass loss can be imagined for forests older than 20 years, but the regression statistics were not convincing (see Tab. S3: regression model types 2 – 6). The relationships are strongly influenced

by forest age and tree mortality intensity. The linear regression models using only one proxy variable already show high significance (Tab. S3: Model types 2 - 6: p-value < 0.01). For instance, the linear model of forest height correlates positively with biomass loss (Tab. S3: Model type 2), which is plausible since forest height is an indicator of the successional stage (see Fig. 3.d). As forest height increased, more tall trees of climax species died, resulting in proportionally more deadwood biomass compared to forests in early successional stages, in which many pioneer species with lower biomass were present and the forest





height was lower (see Fig. 3.d; Fig. S2). Also, the linear model with LAI as proxy variable showed high significance (Table
S3: model type 3), yet LAI was positively correlated with biomass loss, which seems implausible. The LAI is an indicator of
disturbance intensity due to tree mortality. As tree mortality increased, the LAI and mean forest age decreased (Fig. 3.c; Fig.
S3.b). This suggests that model type 3 still includes effects that are not explained by LAI alone, making the expected negative
correlation of LAI with biomass loss a false positive.


**Figure 5: Dependence of biomass loss from single attributes (a) biomass, (b) LAI, (c) forest height versus (d) the multiple forest attributes LAI, forest height of simulated forest states, including all tree mortality scenarios. Each dot represents a terra firme forest stand with a unique forest structure (i.e., tree size distribution and functional species composition). The colours of the dots show the**
**mean forest ages characterising the successional states (ODM: organic dry matter).**





For forests older than 20 years, Fig. 5.d illustrates a three-dimensional relationship between LAI, forest height, and biomass loss. Only when combined in a multiple linear regression model did the LAI (L) and forest height (H) explain the biomass loss of forests at different successional states well ($R^2$ = 0. 94672, RMSE = 0. 00999, p-value $\leq$ 0.01; Fig. 6; Tab. S3):

$$m_{AGB} = 0.004166 \cdot H - 0.030614 \cdot L + \varepsilon. \qquad (7)$$

The LAI negatively influenced biomass loss, whereas forest height is positively correlated with biomass loss. The obtained residuals were normally distributed around the expected value (E($m_{AGB}$) = 0.0; Fig. S5.b), and depending on the estimated biomass loss rates, the residuals were homoscedastic with almost no trend (Fig. S5.c).

The one-to-one comparison of biomass loss rates for the simulated forest stands estimated by the multiple linear regression model (cf. eq. 7) versus those simulated within the dynamic forest model fit well. However, the fitted multiple linear regression

model cannot describe the biomass loss of all forest stands. There is a small number of stands with simulated low but predicted high biomass loss. These are forests of simulated age between 0 – 20 years which are also accompanied by relatively low values of LAI and forest height (cf. Fig. 5.d and Fig. S6). Other forest attributes, such as GPP and NPP, were not included in the multiple linear regression model because they did not improve the estimation of the biomass loss substantially (Tab. S3).

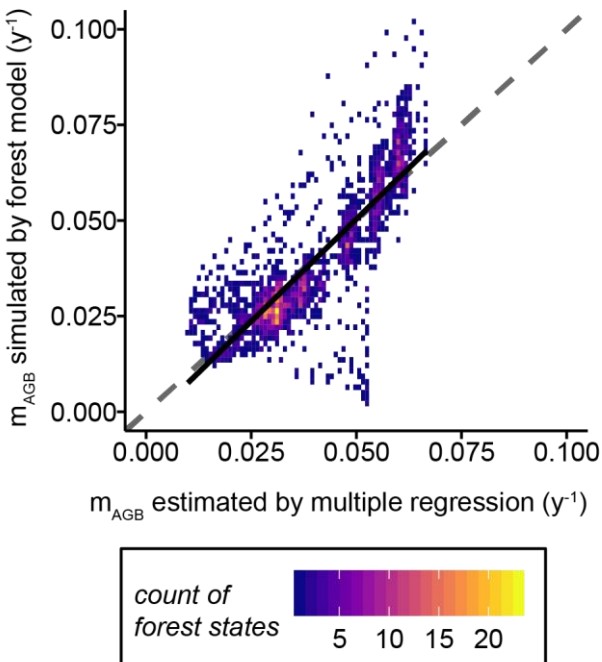

**Figure 6: One-to-one density plot of biomass loss rates simulated by the dynamic forest model versus biomass loss estimated using a multiple linear regression model, with the forest height and leaf area index as proxy variables (eq. 7, Tab. S3). The dashed line shows the line of perfect fit. Each dot represents a forest stand with a unique forest structure (i.e., tree size distribution and functional species composition) while the colours represent the density distributions of the combinations. The black solid line indicates the mean deviations of the biomass loss simulated with the forest model from the estimated ones ($m_{AGB,DFM}$ = 1.0688 · $m_{AGB,LM}$ − 0.0031**

**+ ε, $R^2$ = 0.6905, RMSE = 0.0099, p-value < 0.01). ($m_{AGB}$: biomass loss; DFM: dynamic forest model; LM: multiple linear regression model).**





### 3.3 Estimation of biomass loss from remote sensing by deriving a sample map

By combining simulated forest states with the maps of LAI and forest height obtained via remote sensing (Myneni et al., 2015; Simard et al., 2011), we derived a biomass loss map for terra firme forests of French Guiana (Fig. 7). Based on this sample

map, we obtained a mean biomass loss rate of 0.032 y$^{-1}$ (standard deviation of 0.01 y$^{-1}$). The values of biomass loss varied among regions, with higher values in the southern part of the country and lower rates in the northern part of the country. The highest biomass loss rates can be observed in the centre and at the southwestern country borders (m$_{AGB}$ > 0.06). Such high values resulted from a combination of tall forest height together with low LAI values (Fig. S6). In the region surrounding the Paracou and Nourage sites, the biomass loss rates had values of 0.011 y$^{-1}$ and 0.019 y$^{-1}$, respectively, which agree well with the

mean biomass loss rates we derived from the empirical data of Brienen et al.'s (2015) study (m$_{AGB,Par}$ = 0.011 y$^{-1}$, sd$_{Par}$ = 0.127 y$^{-1}$ ; m$_{AGB,Nou}$ = 0.015 y$^{-1}$, sd$_{Nou}$ = 0.027 y$^{-1}$; Fig. 7.c). The sensitivity analysis revealed the dependence of the mapped biomass loss on the quality of the input data (Fig. 8). The sensitivity is moderate (i.e., $\Delta$m$_{AGB}$ is small) if the LAI and forest height change uniformly by a certain amount. If the changes in the LAI and forest height are contrary, the sensitivity of the mapped biomass loss is high (i.e., $\Delta$m$_{AGB}$ is large), though we assume that a contrarian change in the input data rarely occurs.

Considering forests at all successional states, we derived another sample map for biomass residence time (Fig. S12) calculated from the reciprocal of biomass loss (eq. 5).

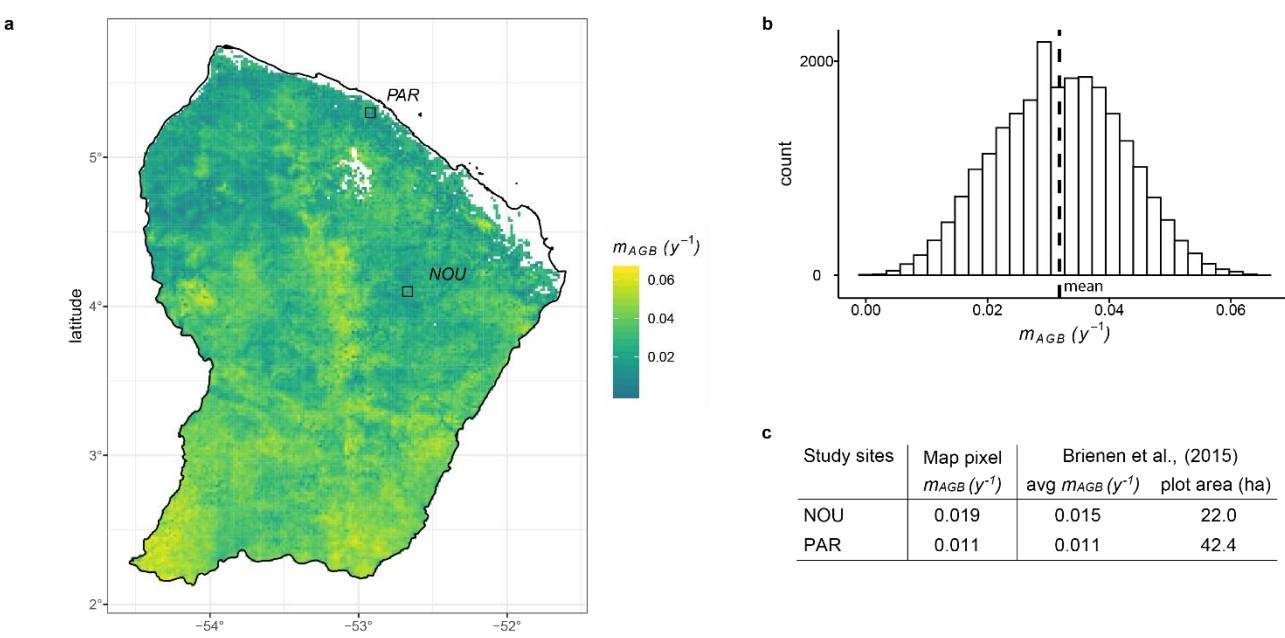

**Figure 7: (a) Map of biomass loss for terra firme forests in French Guiana (~ 2 km resolution) and (b) its histogram. The dashed line in b) indicates the estimated country-wide mean (3.2% with a standard deviation of 1.0%). The black squares in the map show the**
**locations of forest plots at Paracou (PAR) and Nourage (NOU), fro which census-data was used to compare estimated and field-based biomass loss values. (c) The census data originate from Brienen et al., (2015). For further results about the uncertainty analysis performed, see Fig. S7, Fig. 8, Fig S10, and Fig. S11.**





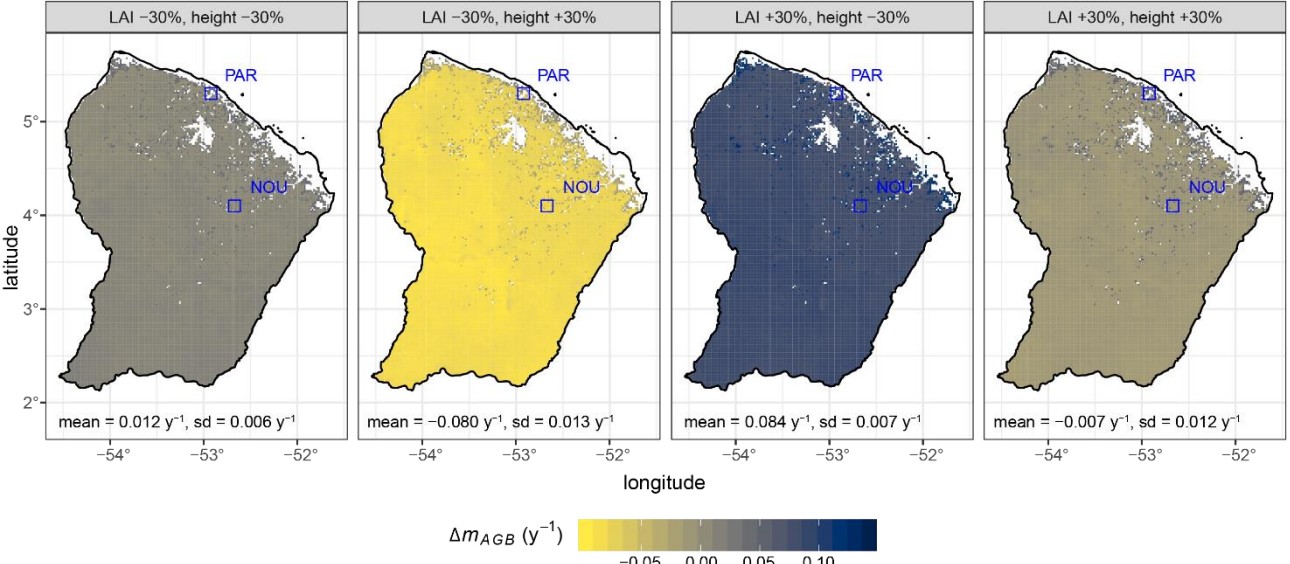

**Figure 8: Sensitivity analysis for the mapped biomass loss of terra firme forests in French Guiana (cf. Fig. 7.a). The values for the input maps of leaf area index (LAI) and forest height were each changed by ±30% from their original values (cf. Fig. S4) and then pair-wise combined to all possible combinations. Δm_AGB represents the variation in biomass loss rates given 30% variation in the input variables. The blue squares show the locations of forest plots at Paracou (PAR) and Nourage (NOU). For each uncertainty map, we calculated the county-wide means and standard deviations (sd).**

## 4 Discussion

### 4.1 Mechanism of tropical forests in dealing with the increasing intensity of tree mortality

In this study, we analysed dynamics in tropical forests in relation to tree mortality. We demonstrated that most of the analysed forest stand attributes (biomass, forest height, LAI, GPP, biomass loss, QMD) had specific responses during succession. Moreover, we showed that biomass loss is strongly affected by succession dynamics as well as by the tree mortality intensity.

The period until the stand's equilibrium was reached differed in duration among each simulation scenario. Additionally, the mean turnover time of biomass, i.e., the reciprocal value of biomass loss (eq. 5), varied considerably.

There were multiple reasons for the unique succession patterns of each forest attribute. Succession dynamics are influenced by assimilation rates (e.g., photosynthesis rates, light requirements) and physiognomic characteristics (e.g., maximum stem diameter increment rates, maximum heights, and wood densities), both of which are specific to each species group (Hiltner et

al., 2018). Functional traits are crucial in simulations of the succession dynamics in forests because they determine the competitiveness of species groups (Fischer et al., 2018; Rüger et al., 2020).

The relationship between successional stages and tree mortality has been investigated in empirical studies to estimate mortality in tropical forests (Aubry-Kientz et al., 2013a; Chambers et al., 2013; Doughty et al., 2015b; Holzwarth et al., 2013). Aubry-



Kientz et al. (2013) introduced a method that estimated the tree mortality probability of terra firme forests at Paracou. Similar
to our results, they found that the tree mortality probability depends on the successional stages of the forests as well as on the
functional traits of species, such as the specific leaf area, wood density, stem diameter increment, and potential height.

Interestingly, we observed similar NPP values at different tree mortality levels for forests in equilibrium. Erb et al. (2016)
argued that the NPP of vegetation is effectively independent of the tree mortality intensity, which is confirmed by our results.
The observed stability of NPP under different disturbance regimes can be explained by shifts within the functional species
compositions and tree size distributions. Pioneer species, which typically have lower wood densities (Chave et al., 2009; Zanne
et al., 2009) and lower potential heights than those of slow-growing climax and intermediate species (Hiltner et al., 2018),
store less carbon in their living biomasses. Since pioneer species grow faster, they can bind as much carbon per time as slow-
growing climax species. Therefore, at the forest stand level, higher tree mortality levels result in similar NPP values as those
observed with lower tree mortality levels, although the individual trees show different growth behaviours. Our simulation
results show how the carbon storage of forests in equilibrium changes across different levels of tree mortality intensity, despite
constant levels of NPP. Instead, our findings indicate that carbon storage depends on the functional species composition. At
high tree mortality rates (e.g., a high-impact scenario), more pioneer trees of a younger age were present in the forest stands.
Thus, to achieve a high forest carbon storage capacity, there is a trade-off between large, old, and less productive trees (e.g.,
climax species) and smaller, younger, and more productive trees (e.g., pioneer species).

**4.2    Performance of the regression model for estimating biomass loss**

One of the main findings of this study is that the simulated biomass loss rates of terra firme forests can be estimated using
multiple linear relationships among several forest attributes. The premise was that all forest attributes used could be provided
by remote sensing and could give information about the forest structure and productivity. We recognized the relationship
between biomass loss to LAI and forest height when fitting many different statistical models with different simulated forest
attributes (Tab. S3). If tree mortality rates increased, this led to a higher biomass loss. However, it is impossible to directly
infer tree mortality rates from biomass loss rates because forest structural state variables differed for each simulated forest
stand, depending on its successional stage (Bohn and Huth, 2017; Rödig et al., 2017). For example, the stem number
distribution of dying trees is not evenly distributed across tree size classes (Aubry-Kientz et al., 2013b; Holzwarth et al., 2013;
Muller-Landau et al., 2006; Rowland et al., 2015).

In our approach for identifying appropriate forest attributes to infer biomass loss, we considered results from empirical studies
that have investigated tree mortality in tropical forests (Aubry-Kientz et al., 2013b; Esquivel-Muelbert et al., 2019; Stovall et
al., 2019). Esquivel-Muelbert et al. (2020) investigated the tree mortality of the Amazon by using empirical data to show that
stem diameter growth rate and tree size are strong predictors. Fast-growing species with low wood densities are at a higher
risk of mortality, whereas the effect of tree size varies. Aubry-Kientz et al. (2013) used functional traits, such as potential tree
height and specific leaf area, to estimate the probability of tree mortality. Based on large-scale remote sensing observations,





tree height was identified as an important predictor of tree mortality during drought, with large trees having twice the mortality rate of small trees, while environmental drivers (i.e., temperature, soil water, and competition) controlled the intensity of the height-mortality relationship (Stovall et al., 2019). The results of these studies underline the importance of productivity (e.g., increment rates and tree size), biomass, and functional characteristics (e.g., wood densities, potential stem diameter increment rates, leaf areas, and potential tree heights) of trees or tree species in the context of tree mortality. In our forest model, such characteristics are included in the derivation of tree mortality rates of specific PFTs (cf. Tab. S1, S2). In forest gap models, forest structural state variables, such as the stem number distribution, tree size distribution, and functional species composition, of the dying trees emerge, rather than get specified as input parameters (Botkin et al., 1972; Bugmann, 2001; Shugart, 2002). This fact is a useful model behaviour for estimating the biomass loss of simulated forest stands and, moreover, holds true for the derivation of other forest attributes, which we considered when fitting our regression model. Besides the LAI and forest height, we tested GPP, NPP, and biomass as proxy variables for biomass loss. On the forest stand level, however, these variables did not improve the performance of the multiple linear regression model substantially. These results suggest using forest height and LAI as proxy variables to estimate the biomass loss of forest stands. Despite the simplicity of the multiple linear regression model, meaning that we included only two proxy variables, its statistical performance proved to be robust (cf. eq. 6; Tab. S3, Fig. S5, Fig. S9, and Tab. S3). Thus, it was possible to derive biomass loss from LAI and forest height of simulated data for forests in different successional states. It was important that the signs of the regression coefficients of our linear model plausibly reflected the relationships that were observed in the field. In the regression model, forest height was directly proportional, and LAI was indirectly proportional to the biomass loss of the forest stands. For example, tall forests with low LAI values resulted in high biomass loss rates (cf. Fig. S6).

Using a forest model to derive the relationships among different forest attributes has several advantages. First, the simulated LAI and forest height data were generated mechanistically, integrating a broad spectrum of information about forest dynamics and successional states emerging from different physiological processes. This can lead to a lower level of noise in the simulation data compared to that in the observed field data. Nevertheless, forest models also include stochastic processes, including tree mortality and establishment (Bugmann, 2001; Fischer et al., 2016; Shugart, 2002). By using plant functional types to simulate forest dynamics, we reduced the possible uncertainties in species traits. Simplifications allow for a transferability of the regression analysis to forests with similar characteristics and succession states. These simplifications also enabled the estimation of the biomass loss rates of terra firme forests across the entirety of French Guiana. With the approach pursued here, it is possible to derive regression models for estimating biomass loss in other locations worldwide. Forest model simulation results contain structural information about the conditions of forests in different successional states, allowing the data to be used as training data for the development of statistical regression models. Whether LAI and forest height are also suitable as proxy variables of the biomass loss rates of other forest types remains to be investigated.



### 4.3    Mapping of biomass loss of terra firme forests in French Guiana

We combined remote sensing maps of forest height (Simard et al., 2011) and LAI (Myneni et al., 2015) with forest modelling to derive a sample map of biomass loss in French Guiana. In doing so, we presented an innovative approach for estimating

biomass loss in tropical forests. A comparison of estimated biomass loss with census-based values for two sites showed reasonable similarity. In another comparison of biomass loss values obtained for French Guiana with census-based values for the entire Guiana Shield (i.e., French Guiana, Suriname, Guyana, northern Brazil, eastern Venezuela; Johnson et al., 2016), our estimate is about 50% higher, though it is noteworthy that Johnson et al. (2016) estimated biomass loss for the entire Guiana Shield, with higher values on average in French Guiana. Capabilities for improved projections of biomass losses are

indispensable in the context of improved estimates of the role of tropical forests in the global carbon cycle (Anderegg et al., 2020; Friedlingstein et al., 2019; Friend et al., 2007; IPCC, 2014). Remote sensing by airborne and satellite-based instruments provides large-scale data on forests, such as the forest height (Simard et al., 2011) and LAI (Myneni et al., 2015). However, remote sensors can record measurements only at certain time points; hence, the successional stages of forest variables are uncertain in remotely sensed data. Such forest dynamics can be simulated by individual-based, dynamic forest models. A

combination of remote sensing data and forest models therefore has the potential to improve predictions of large-scale ecosystem dynamics (Plummer, 2000; Shugart et al., 2015).

Forests can be in different successional stages due to disturbances that influence forest height and LAI (Dubayah et al., 2010; Kim et al., 2017). In the forest height and LAI maps, disturbed regions can be detected visually (cf. Fig. S4); these regions have been identified as disturbed areas in other studies (Asner and Alencar, 2010; Piponiot et al., 2016a; Stach et al., 2009).

Such areas include disturbed areas in the flood plains of lakes and rivers, along the coast, near roads and settlements, and in the secondary forests of French Guiana, where the forest height and the crown coverage is, on average, lower than that in primary forests (Piponiot et al., 2016a; Stach et al., 2009; forest height map from Simard et al. (2011) in Fig. S4).

Remotely sensed products often include uncertainties. In this study, we demonstrated the sensitivity of the sample biomass loss map to variations in the LAI and forest height maps. Accuracy of the input remote sensing data is beneficial. Balanced

deviations of LAI and forest height (e.g., LAI +30% and H + 30%) result in smaller deviations in biomass loss than opposite deviations (e.g., LAI +30% and H -30%). However, small-scale fluctuations in the LAI (e.g., on the individual tree level) were not captured due to the coarse resolution of the MODIS data (500 m). However, by using an individual-based dynamic forest model, the small-scale processes that manifest as variations in the LAI and forest height are accounted for in the simulations. Because simulated forest structures were the basis upon which the regression model was derived, the multiple linear regression

model (eq. 7) accounts for successional states and small-scale dynamics of forests. It is plausible that the LAI alone is not representative of the forest successional state. Therefore, we analysed forest succession (resulting from model-inherent processes), taking species diversity and interactions between trees into account, and we linked this information to the remote sensing products. Such approaches have already been successfully carried out in several studies (Rödig et al., 2017, 2018b,





2019; Shugart et al., 2015). We obtained a good proxy for the successional state and, thus, for biomass loss, only when forest
height was also included in the analysis.

### 4.4    Introduction of an alternative method for estimating biomass turnover time

Information on the carbon balance of forests is important for quantifying the biomass accumulation rates of trees. Various
studies have estimated the turnover time of biomass, which we defined here as the reciprocal value of biomass loss, in forests
worldwide (Carvalhais et al., 2014; Erb et al., 2016; Pugh et al., 2019). Carvalhais et al. (2014) were the first to estimate
biomass turnover times for forests in equilibrium from biomass and GPP (cf. eq. 4: $\tau = AGB_{total} \cdot NPP^{-1}$). For the French
Guiana region, the authors estimated biomass turnover times of approximately 20 to 40 years and discussed that disturbances
can shorten the biomass turnover time by increasing biomass loss rates. Our study quantifies the extent to which tree mortality
leads to high biomass loss and thus to short biomass turnover times.

Erb et al. (2016) observed decreases caused by land use in the biomass turnover time. They found turnover times of 20 to 30
years for the French Guiana region, which are similar to our results (Fig. 4.c). Pugh et al. (2019) showed that stand-replacing
disturbances also shortened the biomass turnover times. We found that the biomass turnover time is strongly affected by
succession dynamics and tree mortality intensities. For our full simulation data set, we found a mean biomass turnover time of
32 years. We derived an alternative framework to estimate the turnover time from biomass loss. This framework allows both
turnover time and biomass loss to be modelled in a simple way, considering succession dynamics and disturbances due to tree
mortality. This method can be applied to forests in equilibrium and to forests in which the early stages of succession are
emerging due to disturbances and logging.

### 4.5    Outlook

Our simulation results revealed complex relationships between tree mortality and biomass loss. The growth stage of a tree
evidently has an effect on tree mortality, which often results in a U-shaped relationship of tree mortality as a function of the
tree size distribution in a forest (Aubry-Kientz et al., 2013b; Holzwarth et al., 2013; Muller-Landau et al., 2006). With regard
to tree age, it is more likely that the youngest and oldest trees will die (Aubry-Kientz et al., 2013b; Rüger et al., 2011) due to
intense competition for light and space between the juvenile trees in the understorey and the senescence of the old trees in the
canopy layer. Such mortality processes are often represented in forest models (Bugmann et al., 2019). Although empirical
mortality algorithms which mechanistically describe the main causes of tree mortality and their effects on entire ecosystems
(e.g., self-thinning, dying of trees by crushing, and growth-dependent mortality) have already been developed, other causes of
tree mortality with unclear signals are often summarised as stochastic processes (Bugmann et al., 2019; Hülsmann et al., 2017,
2018). In our study, biomass loss at the stand level arose from different mortality processes that occurred at the tree level
(competition due to crowding, dying of other trees by crushing, growth dependency, gap formation, and stochastic tree





mortality). Compared to the U-shaped tree mortality distribution, the biomass loss rates of a forest stand depended in more
complex ways on the functional species composition and the levels of carbon fluxes (GPP and NPP).

In our study, the effects of disturbances were represented in a simplified manner by modifying the mortality rates of trees. We
analysed the effects of permanently increasing the tree mortality rates in the studied forests. However, it is also necessary to
consider the effects of discrete or continuously changing disturbance patterns (e.g., Barlow et al., 2003; Brando et al., 2014;
Chambers et al., 2009, 2013; Doughty et al., 2015a; Holzwarth et al., 2013; Magnabosco Marra et al., 2016; Marra et al., 2014;
McDowell et al., 2018; Negrón-Juárez et al., 2010, 2017; Nepstad et al., 2007b; Phillips and Brienen, 2017; Slik et al., 2010;
Stovall et al., 2019; Wright et al., 2015). The impacts of single, discrete disturbance events (e.g., selective logging) on the
dynamics of terra firme forests were studied by Hiltner et al. (2018). A follow-up study investigated the impacts of repeated
logging events under continuously changing air temperatures and precipitation rates (Hiltner et al., 2021).

It was also found that the temporal patterns of establishing trees can change after disturbances such as modifications to the
seed mortality of specific tree species, as such changes influence the competitive processes of trees within communities (Dantas
de Paula et al., 2018). Here, we did not consider the influences of tree mortality intensities on establishment processes, though
this factor should be considered in future studies.

Regarding the mapping of the biomass loss rates in French Guiana, there are three important aspects. First, it is important to
verify the quality of the forest model parameterization with field data as was done for biomass loss in this study and by Hiltner
et al. (2018; 2021), who analysed biomass dynamics, tree size distribution, and functional species composition by comparing
model results with data from forest inventories of the Paracou study site. Second, a multiple linear regression model predicting
biomass loss can be valid only for a certain type of forest. In mapping biomass loss at the country level, we assumed the
predominance of a similar type of forest, the terra firme forests in French Guiana (Hammond, 2005). For this forest type, Stach
et al. (2009) calculated a forest cover of 95% of the country's land area. Third, site parameters across entire landscapes can be
heterogeneous, affecting forest dynamics and structure. Various studies demonstrated that natural environmental factors, such
as soil properties (Rödig et al., 2017; Soong et al., 2020), relief (Guitet et al., 2018), and climatic variations (Rödig et al., 2017;
Wagner et al., 2012), as well as the logging history (Hiltner et al., 2018; Piponiot et al., 2016b, 2019), can affect the succession
dynamics and states of tropical forests. In this study, such spatially heterogeneous environmental influences on forest dynamics
in terra firme forests are indirectly considered in the forest model and the regression model via stochastic tree mortality. In
further investigations, it is recommended that climatic and topographic effects or short-term disturbance events be implemented
to further improve the approach developed here.

## 5   Conclusions

Here, we developed a framework for estimating biomass loss in tropical forests. We analysed the effects of tree mortality
intensity and its relation to forest productivity, forest structure, and biomass, based on the example of terra firme forests in
French Guiana. By quantifying such effects through simulation experiments, it was possible to derive complex relationships between biomass loss and other forest attributes. Our approach revealed the strong influences of the succession states and tree mortality intensity on the biomass loss of forests.

We also linked individual-based forest modelling with remote sensing so that an estimation of biomass loss due to tree mortality was feasible. The resulting sample map of biomass loss indicated that biomass is dying at a faster rate in the central, southern,

and eastern regions than in the northern parts of French Guiana. The forest areas in the north and northeast are used for timber production, agricultural activities, and housing (Bovolo et al., 2018; Stach et al., 2009), whereas the forest areas in the south are predominantly natural rainforests (Hammond, 2005).

The approach we developed here can be easily transferred to other forest biomes (e.g., boreal and temperate forests) using forest models that capture biome-specific forest dynamics and available remote sensing products. Estimating the

spatiotemporal distribution of forest biomass loss has recently been identified as particularly relevant for the monitoring of mortality hotspots (Hartmann et al., 2018). Moreover, improved estimations of the turnover times of carbon in forest stands have been made possible so that uncertainties in the global carbon cycle (Friend et al., 2014) can be reduced.

**Competing Interest**

The authors declare no conflict of interest. The funders had no role in the design of the study, in the collection, analyses, or

interpretation of data, in the writing of the manuscript, and in the decision to publish the results.

**Acknowledgements**

We would like to sincerely thank *Prof. Dr. Achim Bräuning, Prof. Dr. Harald Bugmann,* and *Dr. Nuno Carvalhais* for fruitful discussions of the simulation results and the manuscript. U.H. would like to thank *A. Keberer* for his remarkable assistance. The authors would like to thank both the reviewers, *Dr. Thomas Pugh* and the anonymous second reviewer, and the editor,

*Prof. Dr. Martin de Kauwe*, for their constructive comments during the revision process of the manuscript. U.H. was funded by the *German Federal Environmental Foundation – DBU* [AZ 20015/398] and the programme '*Realization of Equal Opportunities for Women in Research and Teaching' – FFL* of Friedrich-Alexander-University Erlangen-Nuremberg.

**Author's contributions**

U.H., A.H., and R.F. conceived and designed the experiments; U.H. acquired and managed the data; U.H. performed the

simulations; U.H., A.H., and R.F. contributed to analysis and discussion; U.H. wrote the manuscript; U.H., A.H., and R.F. reviewed the manuscript.





**Data availability**

The FORMIND parameterization (Hiltner et al., 2018) and the source code of FORMIND can be downloaded for free on the website www.formind.org. We included a comprehensive description of the model parameters used here in the supplemental

material (Tab. S1 and Tab. S2). The biomass loss map of French Guiana is freely available as online attachment. The input data of the MCD15A2H Version 6 MODIS for the LAI and the forest height map can be downloaded for free (Myneni et al., 2015; Simard et al., 2011)

**Competing Interest**

The authors declare no conflict of interest. The funders had no role in the design of the study, in the collection, analyses, or

interpretation of data, in the writing of the manuscript, and in the decision to publish the results.

**Acknowledgements**

We would like to sincerely thank *Dr. Nuno Carvalhais* and *Prof. Dr. Achim Bräuning* for fruitful discussions of the simulation results and the manuscript. U.H. would like to thank *A. Keberer* for his remarkable assistance. The authors would like to thank the reviewers, *Dr. Thomas Pugh* and the anonymous second reviewer, for their constructive comments during the revision

process of the manuscript. U.H. was funded by the *German Federal Environmental Foundation – DBU* [AZ 20015/398] and the programme '*Realization of Equal Opportunities for Women in Research and Teaching' – FFL* of Friedrich-Alexander-University Erlangen-Nuremberg.





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
