# Peer review of "Importance of the forest state in estimating biomass losses from tropical forests: combining dynamic forest models and remote sensing"

_Biogeosciences, 2021_

## Author Comment (AC1)

**Reviewer comments 1 (https://doi.org/10.5194/bg-2021-195-RC1)**

**"**Importance of the forest state in estimating biomass losses from tropical forests: combining dynamic forest models and remote sensing**"** uses an individual-based model to predict how forest structural attributes (LAI, forest height, others) and carbon fluxes (GPP, NPP, rates of biomass loss) are associated with different levels of tree mortality. The model-derived relationship between LAI, forest height, and biomass loss is applied to remote sensing-derived LAI and forest height data across French Guiana to estimate biomass loss across the entire country.

I think that this manuscript is clearly written and makes a compelling argument for the methodological approach (combining individual based models with remote sensing data to investigate carbon fluxes). In this submission, the authors have added some analyses in response to feedback from previous reviewers. However, I still share some concerns that were raised in the initial submission, and I think that further additional information would help clarify whether it is appropriate to apply the model-derived biomass loss regression to characterize variation in biomass loss across all French Guiana. Here are some main points for consideration:

Thank you very much for your helpful, constructive comments. Below you will find our replies to your comments (highlighted in blue). The line numbers given in our replies refer to the ones in the manuscript.

Note that we changed the terms 'biomass loss' to 'biomass loss rate' and 'tree mortality rate' or 'tree mortality intensity' to 'stem mortality rate' (proposed by Reviewer 2).

1. **A)** I agree with previous comments that this analysis is potentially limited by the field data used for model parameterization, particularly given that the plot has considerably lower biomass loss than the predicted country-wide average. Is the parameterization really representative enough of the whole region? I think that the additional analysis with "altered productivity rates" is meant to address this, but it is difficult to determine whether this analysis is sufficient without additional details. **B)** What does "photosynthesis intensity" mean, in Figure S8? What other model parameters might, plausibly, vary over the study region? From Table S2 I would guess that management parameters (especially give the main conclusions in line 535-537), site-specific climate, and potentially geometric terms could vary (but I'm not an expert in variation within French Guiana). **C)** I don't think that it will be possible to completely resolve this in the study, and I don't think that is necessarily needed for this manuscript to be useful/interesting. However, I think that it would be appropriate to include a more explicit discussion of the limitations of this study. Some of these issues are briefly mentioned at the end of the discussion (Lines 513-526), but I feel that section downplays —rather than acknowledges— the potential limitations of applying results from one model parameterization to all of French Guiana.

Thank you for the comment.

Reply to A: By varying the stem mortality rate and the photosynthetic rate over a broad range, we generated a large dataset of forest stands that has been explored in detail. This artificial dataset of forest dynamics covers a wide range of possible forest states such as the variability in tree species composition, successional state, and tree size distribution. We assume that we can use it to partially cover almost every state of forest stands in French Guiana (so-called forest factory approach, see Bohn et al. 2017). We will amend the text (e.g., in the Abstract, Introduction, and Methods section) to make this point clear. We would like to emphasise that the country-wide biomass loss map has been included in the manuscript as a possible application example for the purpose of showing what is possible with such a model-derived artificial dataset. We will point this out in more detail. In perspective, it would be important to validate such maps with more field data (currently not available to us).

Reply to B: To generate variability in photosynthetic rate, we varied the model parameter 'maximum photosynthetic rate' of the light response curve (Tab. S2). We will add this explanation to Fig S8.

Tab. S2 incorrectly stated the effect of forest management was included in the current simulations. This is not correct and will therefore be deleted from Tab. S2. However, silvicultural interventions shape forest structures, and thus, forest states locally. It would be interesting to consider them in follow-up studies to provide additional information when fitting statistical models to estimate biomass losses regionally. We will mention this point in the discussion.

Reply to C: Thank you for your input. The following further factors can vary on a regional scale and their effects can be considered by adjusting model parameters:
1) Forest management and fire can be simulated.
2) Effects of weather variables such as temperature, rainfall variability, and solar radiation can be considered.
3) Relationships describing tree geometry can vary in space and time.

We will ensure in the revision that the potential limitations due to our assumptions of the dynamic forest model regarding site-specific environmental factors and the representativeness of the model parameterisation for the entire country are discussed in more detail so that they are no longer perceived by the reader as downplaying (e.g., lines 513–526).

2. I think that the sensitivity analysis to remotely sensed LAI and height data is a valuable addition to this revision, and I appreciate that the authors added it in response to previous comments. However, I have some concerns about how this analysis was performed. Why were constant values of +/- 30% chosen? Do the original data sources give estimates of uncertainty associated with these data products? I think it is overly simplistic to assume that the data product would be off by a consistent factor across the entire country—a more useful analysis would be simulating heterogenous variation in LAI and height estimates across the country, perhaps using a Monte Carlo approach. Given that these data are input to a simple linear relationship, I don't think this change would be computationally unreasonable.

Thank you for pointing this out. We assumed the homogeneous variations of ±30% in the input variables (LAI and forest height) as a "worst-case" scenario. We follow the reviewer's suggestion and will add a sensitivity analysis as suggested (sampling random numbers per pixel as factors for the LAI and forest height variations).

3. A) In addition to the differences in spatial scale between the datasets mentioned by previous reviewers, I am concerned that "forest height" as estimated in the model and quantified at the 1 km scale are not interchangeable. Simard et al. (2011) use the mean height of the 3 tallest trees to validate GLAS data at the footprint level, but not to validate the gridded 1-km data product, which tends to be shorter and less variable (Table 2 in Simard et al.). The gridded product is based on a biome-level Random Forest model using other ancillary data (tree cover %, precipitation, elevation, temperature, protection status), so variation in 1-km forest height across French Guiana doesn't necessarily reflect "measured" variation in forest structure, but instead predicted variation based on biome-level correlations with other factors. I understand that the authors might not be able to do much about this limitation—but I do think that it deserves at stronger caveat in the discussion section, at least. B) What factors (tree cover and/or climate?) do you think are most important for driving the height and/or LAI maps, and subsequently the predictions of biomass loss?

Thank you very much for the comment.

Reply to A: We agree with the reviewer. As noted, we unfortunately cannot eliminate the limitation regarding the values of forest height derived by Simard et al. (2011). We will add the aspects the reviewer mentioned in the discussion on the limitations of the remote sensing data (lines 425 fol.).

Reply to B: We think that both factors have an impact on forest height, LAI, and biomass loss. For example, drought, uprooting due to storms and flooding, fire, insect calamity, forest management, etc. may be possible drivers of variability in the LAI. Forest height can vary due to uprooting from storms and flooding, fire, forest management, etc. Those environmental drivers may also interact with each other, too. The analysis of this question is interesting but is beyond the scope of our study. It should be explored in follow-up studies. We will add text in the discussion of the limitations of our linear regression model, because unfortunately we cannot perform analyses regarding the question in this study.

4. **A)** The methods section claims "No correction factors were required for the extrapolations (see Fig. S7)" for LAI and height data, but Figure S7 shows that much of the country-wide data (perhaps ~25%?) falls outside of the range of simulated values. What information was used to determine that no correction was necessary? **B)** In addition to (or instead of) Figure S7, it would be helpful to have a figure showing the range of data in 2D LAI/forest height parameter space from simulations and from remote sensing. I recommend something like Figure S6, but with a heat map showing the density of remote sensing data in the background, and the simulation trajectories overlaid.

Thank you for the comment.

Reply to A: We agree with the reviewer, the value range of the remotely sensed data is partly outside of the simulated value range (Fig. 7). In the revision, we will test whether harmonising the remote sensing products against the simulated data will improve the estimated biomass loss rates. For example, we will reduce the remotely sensed LAI values following the rationale that no understory trees (DBH < 10 cm) are considered in FORMIND. We will then recompile the biomass loss map.

Reply to B: Very good idea. We will provide a heat map as suggested.

5. It would also be helpful to have some additional details to evaluate how well the multiple linear regression model characterizes the modeled relationship between LAI, forest height, and biomass loss. In figure S5c, it does look like the residuals have an apparent "smile" shape—in particular, the residuals are consistently positive in the range of the only field data included for comparison (0.011-0.015 $y^{-1}$). For example, I would like to see (supplemental) figures showing the relationship between forest height and residuals of the single attribute LAI/biomass loss regression, and vice versa, colored by forest age.

Thank you for the comment. We followed the reviewer's suggestion and created the plots already (Fig. R1). The residuals attributed to the LAI contain no remaining trend for forest stands older than 40 years. The ones attributed to forest height still contain a remaining trend, which is small and age-dependant. So we can prove that forest height and LAI can be used as proxy variables to estimate biomass losses.

We would like to note that to further improve the linear regression model (further minimize the residual's remaining trend), additional proxy variables would need to be included. In this study, we tested various forest attributes available as remote sensing products (see Table S3). We decided to use the simplest possible linear regression model (in terms of the number of proxies) that estimated biomass loss rates best.

We will supplement the analyses (Fig. R1) in the Appendix and describe how we derived the residuals attributed to the forest attributes. Also, we will discuss this aspect in the limitations of the multiple

linear regression model.

[Figure]

*Figure R1: Relationship between the residuals associated with the proxy variables of the LAI (left) and the forest height (right) and the simulated forest attributes. Colours indicate the mean forest age.*

A few more minor line items:

6. Line 24: In "changed the forests' gross…", consider replacing "changed" with a more specific word—increased, decreased?

Thank you. We will reformulate the sentence. Here, higher stem mortality rates led to increased levels of GPP.

7. Lines 144-145: This sentence identifies "extreme climate events, forest fires, wind-throw, and diseases" as possible disturbances relevant to this simulation but I think that all of these would cause disturbances of limited duration. Perhaps other examples, like sustained increased temperature and/or reduced water availability would be more appropriate?

Thanks for the hint. We will refer to sustained elevated temperatures and reduced soil water availability as examples.

8. Line 195: The linear regression model does assume that all forest states are independent, but is this a valid assumption? In the model trajectories, there is clear temporal autocorrelation—one state is used as input to the next state in time, correct?

Thank you for asking. We acknowledged that when fitting linear regression models, it is important that the proxy variables are independent of each other. We tested this using a covariance matrix of the predictors (unpublished results) and we will visualize that in the diagrams accompanying your comment 5. Also, we will rephrase the text to make our point more understandable and add the covariance matrix of the predictors to the supplementary materials.

9. Line 321: It is unclear to me whether forests age 0-20 were included in the multiple linear regression model. This line indicates they were, but from the sentence above (Line 312) I thought they weren't.

Thank you for the comment. We included all simulated years in the analyses (years 0 – 300). We will reword the sentence in line 312 so that there is no misunderstanding.

10. Line 497: Are mortality rates from these different mechanisms available as model output? Obviously, there is a lot in this study already, but I wonder looking at how different mortality modes respond to the uniform increase in "base mortality" could provide more mechanistic predictions into how forests respond to sustained increased mortality.

Thank you for this question. Yes, the model outputs for different modes of mortality do exist, however, we did not explore them in this study because it was not part of the research question. It is a

good suggestion to include them in future studies. In this study, we analyzed the relationships between stem mortality rates with biomass loss rates, GPP, NPP, and biomass stock (see Fig. R2 on the relationship with the biomass loss rate). We will include more details on this issue in the supplementary material and add text to the methods and results sections. We will revise Table 1 to report resulting stem mortality rate and biomass loss rate.

[Figure]

*Figure R2: Biomass loss rates $m_{agb}$ versus stem mortality rates $m_{sn}$ for the simulated forest stands. The dashed lines indicate the 1:1-lines.*

11. Line 507: Are there any results from Hiltner et al. (2021) that could be briefly compared the predictions in this study?

Thank you for the question. If additional effects, such as climate change and forest management, were added to the dynamic forest model's simulations, the reasons of biomass losses could be accurately determined. This would be interesting for further studies where the methodology presented here can be used. We will address this when we revise the discussion (see also reply 1.C)

12. Data availability statement: I think it would be useful to include a supplemental file with the model output used to make Figures 2-6. This would allow others to look more closely at the data without having to learn to run FORMIND.

Thank you. We will publish FORMIND's simulation results and make that clear in the "Data Availability" section.

**Literature**

Bohn, F. J. and Huth, A.: The importance of forest structure to biodiversity–productivity relationships, R. Soc. Open Sci., 4(1), 160521, doi:10.1098/rsos.160521, 2017.

---

## Author Comment (AC2)

**Reviewer comments 2 (https://doi.org/10.5194/bg-2021-195-RC2)**

This paper is a revision of Hiltner et al. (2020, https://doi.org/10.5194/bg-2020-264). As with that manuscript, I find that the current one provides very interesting insights into how biomass mortality rates vary as a function of successional stage, as well as providing a useful upscaling method that uses satellite products to extrapolate to country-scale. The study is clearly motivated, structured and written, and the interpretation of the results appropriately caveated. I see that the authors have addressed most of my comments on their previous manuscript and I only have a few minor points on this one. If they are addressed, then I very much recommend publication.

Thank you very much for your helpful, constructive comments. Below you will find our replies to your comments (highlighted in blue). The line numbers given in our replies refer to the ones in the manuscript.

1. The term "biomass loss" is used throughout, but this is a bit of an ambiguous term as it could refer to either the loss rate or the flux. It's clear from the units that it's the rate, but I also think the clarity of the text would be improved if the term "biomass loss rate" was used instead. Similarly, the settings in Table 1 are described as "tree mortality rate" or "tree mortality intensities", I suggest calling them instead "stem mortality rates", as this clearly differentiates from biomass loss rate (maybe it's only in my head, but "tree mortality" feels to me a more general term). Being very specific in the text about what the prescribed and simulated rates are might help emphasise the point being made in the discussion about one not equalling the other.

Thank you. We will change the terms 'biomass loss' to 'biomass loss rate' and 'tree mortality rate' or 'tree mortality intensity' to 'stem mortality rate' throughout the text (e.g., lines 1, 14, 17, etc.).

2. Given that one of the key take-home messages of the manuscript is how successional stages influence how stem mortality rate links to biomass loss rate, it would be very helpful to quantify stem mortality rate in a way that makes it directly comparable to the simulation results. The values in Table 1 are a simple average of the rates of the 8 PFTs, but the actual stem mortality will be the combination of this and the prevalence of the PFTs. So, a fair comparison of the two rates (start of section 4.2) requires the actual stem mortality realised in the simulations.

Thank you for the suggestion. A scenario was defined by changing the mortality parameters. We prepared a figure illustrating the simulated biomass loss rates versus the realized stem mortality rates of each scenario. We calculated the stem mortality rate ($m_{SN}$) as the ratio of the number of dead trees to the number of trees in a forest stand at each simulation time step. In each tree mortality intensity scenario, the biomass loss rate ($m_{AGB}$) is on average greater than the stem mortality rate (see Fig. 1: panels). We will include this figure in the supplementary material and add text to the methods and results sections. We will also revise Table 1 to report resulting mean stem mortality rate rather than the average of tree mortality parameters.

[Figure]

*Figure 1: Biomass loss rates m_AGB versus stem mortality rates m_SN for all simulated forest stands. The dashed lines indicate the 1:1-lines.*

3. In lines 179-189 it's not entirely clear to me which definition is being used for NPP. Is it woody NPP only, or true NPP (i.e., GPP minus autotrophic respiration)? Clarity on this is important because it influences comparisons of the results for turnover time to others in the literature. For instance, a direct comparison to Erb et al. (2016) (line 479) would only be fair if true NPP is being used, as that is (as far as I can tell) what is used by Erb et al.

Thank you for asking. In our study, NPP of a tree is the difference between gross primary production and autotrophic respiration (can be summed up to stand level). Accordingly, one can compare the values of NPP of both studies. We will add this point to the manuscript and add the equation of our NPP calculation to avoid misunderstandings.

4. Line 294, "relationship between single forest attributes and biomass loss can be imagined… but the regression statistics were not convincing", and line 295, "linear regression models using only one proxy variable already show high significance", seem to contradict one another? In fact, I think the whole text on lines 296-304 is distracting and unnecessary. It describes linear regression results (e.g. LAI and biomass loss) that, whilst statistically significant, a glance at Fig. 5 shows are not useful, because the relationship is clearly non-linear. Can be enough simply to show Fig. 5 and state that?

Thank you very much for the comment. We agree with the reviewer that the wording of the mentioned text is complicated. We will shorten and reword it as suggested (lines 294 – 304).

5. The last sentence of Section 4.4 is, I think, still on shaky ground. By definition, forests in the early stages of succession are not in equilibrium, or anywhere near it. The biomass loss rate is following a pretty clear evolution over time in the first 100 years of succession (Fig. 3a), so a derivation of turnover time based on instantaneous biomass loss rate is going to be misleading. It's not going to represent well how long the carbon being fixed at that moment stays in the system. I think it's reasonable to make the calculation over a large area which incorporates forests in various successional states and where one can reasonably expect that those states are fairly close to dynamic equilibrium (as in the rest of the paragraph). But a forest in the early stages of succession is really a long way from equilibrium. I suggest deleting the last clause of this sentence.

Yes, will be done.

6. Line 27. Is this biomass loss at equilibrium?

These are average values over the entire simulated time (years 0 - 300). We will revise the text.

7. Table 1 caption. "$a^{-1}$" is used here, but normally "$yr^{-1}$" in the rest of the manuscript.

Thank you, we will adjust the unit.

8. Fig. 5 caption. ODM seems to be defined for the first time here but is first used on line 242. Could you also define it when first introduced?

Thank you. We will define ODM at the place where it is used for the first time (line 242).

9. Fig. 8. I'm being really picky here (sorry), but I found this a bit awkward to read because it's a scale that spans zero, but the colours are not centred around zero. Could you maybe centre the grey range on zero?

Yes, we do. Thanks for the hint.

10. Line 378. I think "confirmed" is too strong. Maybe "supported"?

Thank you. We will implement the proposed amendment.

11. Line 498. "death of other trees"

Thank you. We will rename 'dying of other trees' to 'death of other trees'.

---

## Author Response (AR1)

**Reviewer comments 1 (https://doi.org/10.5194/bg-2021-195-RC1)**

"Importance of the forest state in estimating biomass losses from tropical forests: combining dynamic forest models and remote sensing" uses an individual-based model to predict how forest structural attributes (LAI, forest height, others) and carbon fluxes (GPP, NPP, rates of biomass loss) are associated with different levels of tree mortality. The model-derived relationship between LAI, forest height, and biomass loss is applied to remote sensing-derived LAI and forest height data across French Guiana to estimate biomass loss across the entire country.

I think that this manuscript is clearly written and makes a compelling argument for the methodological approach (combining individual based models with remote sensing data to investigate carbon fluxes). In this submission, the authors have added some analyses in response to feedback from previous reviewers. However, I still share some concerns that were raised in the initial submission, and I think that further additional information would help clarify whether it is appropriate to apply the model-derived biomass loss regression to characterize variation in biomass loss across all French Guiana. Here are some main points for consideration:

Thank you very much for your helpful, constructive comments. You will find our replies to your comments below (highlighted in blue). The line numbers given in our replies refer to the ones in the manuscript with changes tracked.

Note that we changed the terms 'biomass loss' to 'biomass loss rate' and 'tree mortality rate' or 'tree mortality intensity' to 'stem mortality rate' (proposed by Reviewer 2).

1. A) I agree with previous comments that this analysis is potentially limited by the field data used for model parameterization, particularly given that the plot has considerably lower biomass loss than the predicted country-wide average. Is the parameterization really representative enough of the whole region? I think that the additional analysis with "altered productivity rates" is meant to address this, but it is difficult to determine whether this analysis is sufficient without additional details. B) What does "photosynthesis intensity" mean, in Figure S8? What other model parameters might, plausibly, vary over the study region? From Table S2 I would guess that management parameters (especially give the main conclusions in line 535-537), site-specific climate, and potentially geometric terms could vary (but I'm not an expert in variation within French Guiana). C) I don't think that it will be possible to completely resolve this in the study, and I don't think that is necessarily needed for this manuscript to be useful/interesting. However, I think that it would be appropriate to include a more explicit discussion of the limitations of this study. Some of these issues are briefly mentioned at the end of the discussion (Lines 513-526), but I feel that section downplays —rather than acknowledges—the potential limitations of applying results from one model parameterization to all of French Guiana.

**Thank you for the comment.**

Reply to A: By varying the stem mortality rate and the photosynthetic rate over a broad range, we generated a large dataset of forest stands that has been explored in detail. This artificial dataset of forest dynamics covers a wide range of possible forest states such as the variability in tree species composition, successional state, and tree size distribution. We assume that we can use it to cover most of the forest states in French Guiana (so-called forest factory approach, see Bohn et al. 2017). We amended the text (e.g., in the Abstract (line 21) and Methods (line 112 fol.) to make this point clear. We would like to emphasise that the country-wide biomass loss map has been included in the manuscript as a possible application example for the purpose of showing what is possible with such a model-derived artificial dataset. In perspective, it would be important to validate such maps with more field data (currently not available to us). We pointed this out in more detail in the Introduction (line 102 fol.) and Discussion (lines 495 fol., 585 fol.).

Reply to B: To generate variability in photosynthetic rates, we varied the model parameter 'maximum photosynthetic rate' of the light response curve (cf. Tab. S2). We added this explanation to the caption of Fig S10.

Tab. S2 incorrectly stated the effect of forest management was included in the current simulations. This is not true and was therefore deleted (cf. Tab. S2). However, silvicultural interventions shape forest structures, and thus, forest states locally. It would be interesting to consider them in follow-up studies to provide additional information when fitting statistical models to estimate biomass loss rates regionally. We mention this point in the discussion (lines 598 fol.).

Reply to C: Thank you for your input. The following further factors can vary on a regional scale and their effects can be considered by adjusting model parameters:

1) Forest management and fire can be simulated.

2) Effects of weather variables such as temperature, rainfall variability, and solar radiation can be considered.

3) Relationships describing tree geometry can vary in space and time.

We included a new section in the revision about potential limitations due to our assumptions of the dynamic forest model regarding site-specific environmental factors and the representativeness of the model parameterisation for the entire country (e.g., lines 595 fol.).

2. I think that the sensitivity analysis to remotely sensed LAI and height data is a valuable addition to this revision, and I appreciate that the authors added it in response to previous comments. However, I have some concerns about how this analysis was performed. Why were constant values of +/- 30% chosen? Do the original data sources give estimates of uncertainty associated with these data products? I think it is overly simplistic to assume that the data product would be off by a consistent factor across the entire country—a more useful analysis would be simulating heterogenous variation in LAI and height estimates across the country, perhaps using a Monte Carlo approach. Given that these data are input to a simple linear relationship, I don't think this change would be computationally unreasonable.

Thank you for pointing this out. We assumed the homogeneous variations of  $\pm 30\%$  in the input variables (LAI and forest height) as a "worst-case" scenario. We followed the reviewer's suggestion and added a sensitivity analysis as suggested (sampling random numbers per pixel as factors for the LAI and forest height variations, see Fig. 7.d).

3. A) In addition to the differences in spatial scale between the datasets mentioned by previous reviewers, I am concerned that "forest height" as estimated in the model and quantified at the 1 km scale are not interchangeable. Simard et al. (2011) use the mean height of the 3 tallest trees to validate GLAS data at the footprint level, but not to validate the gridded 1-km data product, which tends to be shorter and less variable (Table 2 in Simard et al.). The gridded product is based on a biome-level Random Forest model using other ancillary data (tree cover %, precipitation, elevation, temperature, protection status), so variation in 1-km forest height across French Guiana doesn't necessarily reflect "measured" variation in forest structure, but instead predicted variation based on biome-level correlations with other factors. I understand that the authors might not be able to do much about this limitation—but I do think that it deserves at stronger caveat in the discussion section, at least. B) What factors (tree cover and/or climate?) do you think are most important for driving the height and/or LAI maps, and subsequently the predictions of biomass loss?

Thank you very much for the comment.

Reply to A: We agree with the reviewer. As noted, we unfortunately cannot eliminate the limitation regarding the values of forest height derived by Simard et al. (2011). We added the aspects the reviewer mentioned in the discussion on the limitations of the remote sensing data (lines 525 fol.).

Reply to B: We think that both factors have an impact on forest height, LAI, and biomass loss. For example, drought, uprooting due to storms and flooding, fire, insect calamity, forest management, etc. may be possible drivers of variability in the LAI. Forest height can vary due to uprooting from storms and flooding, fire, forest management, etc. Those environmental drivers may also interact with each other, too. The analysis of this question is interesting but is beyond the scope of our study. It should be explored in follow-up studies. We added text in the discussion of this outlook (lines 596 fol.).

4. The methods section claims "No correction factors were required for the extrapolations (see Fig. S7)" for LAI and height data, but Figure S7 shows that much of the country-wide data (perhaps ~25%?) falls outside of the range of simulated values. What information was used to determine that no correction was necessary? In addition to (or instead of) Figure S7, it would be helpful to have a figure showing the range of data in 2D LAI/forest height parameter space from simulations and from remote sensing. I recommend something like Figure S6, but with a heat map showing the density of remote sensing data in the background, and the simulation trajectories overlaid.

**Thank you for the comment.**

We agree with the reviewer, the value range of the remotely sensed data is partly outside of the simulated value range (Fig. S9). In the revision, we corrected the remote sensing products against the simulated data. We reduced the remotely sensed LAI values following the rationale that no understory trees (DBH

Figure R1: Simulated biomass loss rates mAGB versus simulated stem mortality rates mSN for all simulated forest stands. One dot represents a simulated forest stand with a size of 1 ha. The dashed line indicates the 1:1-line.

3. In lines 179-189 it's not entirely clear to me which definition is being used for NPP. Is it woody NPP only, or true NPP (i.e., GPP minus autotrophic respiration)? Clarity on this is important because it influences comparisons of the results for turnover time to others in the literature. For instance, a direct comparison to Erb et al. (2016) (line 479) would only be fair if true NPP is being used, as that is (as far as I can tell) what is used by Erb et al.

Thank you for asking. In our study, NPP is the sum of the NPP values of all trees in a stand. NPP at tree level is the difference between gross primary production and autotrophic respiration (can be summed up to stand level). Accordingly, one can compare the values of NPP of both studies. We added the equation of our NPP calculation to avoid misunderstandings (lines 209 fol.).

4. Line 294, "relationship between single forest attributes and biomass loss can be imagined... but the regression statistics were not convincing", and line 295, "linear regression models using only one proxy variable already show high significance", seem to contradict one another? In fact, I think the whole text on lines 296-304 is distracting and unnecessary. It describes linear regression results (e.g. LAI and biomass loss) that, whilst statistically significant, a glance at Fig. 5 shows are not useful, because the relationship is clearly non-linear. Can be enough simply to show Fig. 5 and state that?

Thank you very much for the comment. We agree with the reviewer that the wording of the mentioned text was complicated. We shortened and reworded it as suggested (lines 333 fol.).

5. The last sentence of Section 4.4 is, I think, still on shaky ground. By definition, forests in the early stages of succession are not in equilibrium, or anywhere near it. The biomass loss rate is following a pretty clear evolution over time in the first 100 years of succession (Fig. 3a), so a derivation of turnover time based on instantaneous biomass loss rate is going to be misleading. It's not going to represent well how long the carbon being fixed at that moment stays in the system. I think it's reasonable to make the calculation over a large area which incorporates forests in various successional states and where one can reasonably expect that those states are fairly close to dynamic equilibrium (as in the rest of the paragraph). But a forest in the early stages of succession is really a long way from equilibrium. I suggest deleting the last clause of this sentence.

Yes, we deleted the sentence (line 547 fol.).

6. Line 27. Is this biomass loss at equilibrium?

Yes. We revised the text (line 28).

7. Table 1 caption. "a-1" is used here, but normally "yr-1" in the rest of the manuscript.

Thank you, we corrected the unit.

8. Fig. 5 caption. ODM seems to be defined for the first time here but is first used on line 242. Could you also define it when first introduced?

Thank you. We defined ODM at the place where it is used for the first time (line 279).

9. Fig. 8. I'm being really picky here (sorry), but I found this a bit awkward to read because it's a scale that spans zero, but the colours are not centred around zero. Could you maybe centre the grey range on zero?

Yes, we did (Fig. 7.d, Fig. S14). Thanks for the hint.

10. Line 378. I think "confirmed" is too strong. Maybe "supported"?

Thank you. We implemented the proposed amendment (line 425).

11. Line 498. "death of other trees"

Thank you. We renamed 'dying of other trees' to 'death of other trees' (lines 557 and 561).

---

## Author Response (AR2)

**Editor comments**

Thanks for your careful revision, which was re-reviewed by referee #1 (the other referee was not available). Referee 1 acknowledges many improvements but, as you can see, still has a number of concerns; the referee is concerned that not all of their initial concerns were appropriately addressed/acknowledged, particularly with respect to regression model performance and the sensitivity analysis.

I have read the revised manuscript, the response to reviewers, and Referee 1's assessment. I agree that much improvement has been made--thank you--and think this manuscript is well on track for final acceptance. Nonetheless, I agree with many of R1's thoughtful comments, and would ask that you consider them carefully, and use them to make (hopefully final) improvements and clarifications. The one exception is with respect to the sensitivity analysis; while ±30% is arbitrary, it still provides a useful outer bound, and improving it (as the referee suggests) is entirely optional. Conversely, please address the referee's regression model comments fully.

Thank you very much for your helpful, constructive comments. As main changes to the manuscript, we thoroughly revised the regression model and tested different model types. We have found a new variant that solves the criticisms of the reviewer. This even leads to a partial improvement of the results and now no correction of the remote sensing data is necessary anymore. We believe that the analysis with the ±30% limits (Fig. 7.d) can already give the reader a useful indication about the sensitivity of the biomass loss rates. We would therefore like to follow your suggestion and leave the sensitivity analysis as it is. We added text to indicate that a calculation of an error propagation with provided uncertainty products could be performed (lines 267 fol.), but this is behind the scope of this study (since the map is only an example application of the presented workflow). For our detailed responses, please see the response letter.

**Reviewer comments 1**

This study uses simulated forest data to explore how mortality rate variation influences the structure and dynamics of tropical forests, and to evaluate whether forest structural attributes can be used to predict biomass loss rates from large scale remote sensing data. This revision appropriately addresses some of my recommendations/questions from the previous submission, and I think that the general approach would be a valuable contribution to the field.

Thank you very much for your helpful comments. As main changes to the manuscript, we thoroughly revised the regression model and the comparisons of forest height and LAI from the remote sensing products with the forest model simulation results. Please find our replies to your comments below (highlighted in blue). The line numbers given in our replies refer to the ones in the manuscript with changes tracked.

With the additional information provided in the supplement, I have some remaining concerns about how well the multiple linear regression model describes variation in the simulated dataset, and about the application of that model to the remote sensing data.

First, as you mention, Figure S7b still seems to show some "smile" effect—there is a negative trend with LAI for LAI ~2.5-3.5, and then a positive trend for LAI > ~3.5. It's a little hard to evaluate how much this matters because the units here are not in biomass loss rates, but this suggests to me that a linear relationship with LAI might not describe the trend well, especially because much of the MODIS LAI data appear to fall in the 3-4 range.

Thank you for the comment. The slight "smile" effect in the residuals of the LAI indicates a non-linear component. Therefore, we tested transformations of the input data for the independent variables (i.e., LAI and forest height) and we tested non-linear models, such as generalized additive models (GAMs; Fig S15). Using a GAM, the trends in the residuals of LAI and forest height could be removed (Fig. S15.c and S15.d).

When looking at the residuals of the fitted biomass loss rates and the frequency distribution of the residuals of the GAM (Fig. S15.b and Fig. S15.e) compared to those of the multiple linear model (Fig. S6.b and S6.c), the GAM shows only slight improvement. Since GAMs are not easy to interpret and to communicate (no closed model equation), we decided to focus on an improved linear regression approach. This improved regression model reduced the overall residual trends (new Figures S6.c and S7).

To address the Reviewer's comment in detail, we added the GAM approach as an alternative to the linear model but present the results only in the Supplements (Fig. S15). We improved the regression model (Eq. 7, Tab. S3) and adjusted all analyses and results based on it accordingly (e.g., Ch. 3.2, Fig. 6, Fig. 7, Fig. S6 - S8, Fig. S11 - S14). Finally, we revised the corresponding text in the Methods (lines 215 fol.) and the Discussion (lines 453fol.).

Second, it appears to me that even after the added correction in this revision, the remotely sensed LAI-height parameter space is not very well described by the simulated data—indicating that the "factory forest" approach assumption that the simulated data "cover most of the forest states in French Guiana" is not met. Figure S9b shows that the simulated data are a highly uneven and incomplete sample of the possible values of LAI/height seen in the country-wide data. For example, forests less than ~35m are common in the remote sensing data but only appear in young forests in the simulated data, and much of the simulated data is still lower LAI than the remote sensing data. Also, LAI and height are indeed correlated in the simulated data (Table S4), but it appears that they are not correlated in the remote sensing data.

Thank you for the comment. We agree with the Reviewer that despite the corrections made to the remote sensing products, it looked like they did partly not match the simulation data so well. We therefore carefully checked once again our approach for the comparison of the two data sets and have been able to correct an inconsistency:

- We now omit pixels of the map for which the regression model (see eq. 7) did not predict biomass loss rates, i.e., we excluded negative biomass loss predictions (s. Fig. 7.a, blank pixels). This was mainly the case for populated coastal and aquatic areas, i.e., this approach eliminated pixels that we assume are predominantly unforested (see e.g., Methods and Discussion lines 252 fol., 488 fol.).

- In our revised comparison, we include only those pixels of the remotely sensed LAI and forest height which are also considered in the final biomass loss map. In Fig. S9 (new), it can be seen that the centroid of the remote sensing products and the simulations are now in good agreement, with only a few combinations deviating (shown in light-grey) and without the need to correct the remote sensing products. Therefore, we reversed the correction of the remote sensing data (lines 241 fol.), which is no longer needed due to the improved regression model (see our reply 1). We updated Figure 7.a showing this biomass loss map and affected analysis (e.g., Fig. S13, Fig. S14, Fig. S12). We added a description in the Methods (lines 246 fol.).

Third, I also think that the uncertainty analysis can/should be further improved. Instead of using the arbitrary +/- 30% range, I recommend propagating the actual uncertainty from the remote sensing data, and from the model with forest attributes. There is an uncertainty product associated with each MODIS LAI measurement (LaiStdDev_500m2)—is there a similar metric associated with the height product? For example, instead of sampling from an arbitrary uniform distribution one time, the value for each pixel could be repeatedly sampled from the actual uncertainty distributions, and delta mAGB could be calculated using the 95th percentile range of resampled values. My apologies for not being clearer with this recommendation in the previous review.

Thank you for the suggestion. This study is mainly about method development for estimating biomass losses from forest attributes and the map serves as a sample application (e.g., see lines 98 fol., line 120, caption of fig. 1, and section header 3.3). We discussed the Reviewer's suggestion and found that an additional sensitivity analysis based on available uncertainty products would be extensive and not needed. We think that the analysis with the ±30% limits (Fig. 7.d) can already give the reader a useful indication about the sensitivity of the biomass loss rates in the sample map. We added text in the Methods (lines 262 fol.) to emphasize more clearly that error propagation could be considered in follow-up studies.

I also have a few specific recommendations that can be easily/quickly addressed (line numbers refer to the revised manuscript without tracked changes):

- Line 30: I recommend changing "and mapped" to "to map" to make clear there isn't a separate remote sensing dataset of mapped biomass loss.

Thank you, done (line 30).

- Lines 142-143: I find the sentence "Our assumption…is that forest stands also have no explicit position" a little unclear. Does this mean that different 1 ha forest stands within the 16-ha forest area don't have explicit position/interact with each other, or something else?

Thank you. The forest stands of 1 ha do not have an explicit position within the landscape. We reformulated the sentence to make our point clearer (lines 174 fol.).

- Line 158: Perhaps reference Table S2 here for difference in the PFTs?

Thank you, done (line 159).

- Line 170: Perhaps reword to "in combination with the subsequent effects on the other models of modeled mortality" or something similar, if that is accurate?

Thank you. We reformulated the sentence (lines 170 fol.)

- Lines 226-227: I recommend describing what data were used to product the forest height estimates (i.e. GLAS).

Thank you, we added the description (line 231 fol.).

- Figure 4: Is this the background mortality rate (consistent with the ranges shown here) or the resulting average stem mortality rate as defined in Table 1?

The x-axis shows the background mortality rate. We revised the axis titles to make this clearer (see Fig. 4)

- Figure 7: The map pixel values in the table (c) have not been updated from the previous draft.

Thank you for the hint. We corrected the numbers in Figure 7.c.

- Line 497: I think the last sentence should the header for the next section, correct?

Yes, thank you. We re-formatted it as header (line 514).

- Line 575: I recommend using wording such as "The resulting sample map of biomass loss predicted…" instead of as "The resulting sample map of biomass loss indicated…".

Thank you, done (line 594).